psychology/behaviour

go/no-go training, action tendencies, approach bias, food, liking, choice

**Author for correspondence:**
Loukia Tzavella
e-mail: tzavellal@cardiff.ac.uk

# Effects of go/no-go training on food-related action tendencies, liking and choice

Loukia Tzavella[1], Natalia S. Lawrence[2],
Katherine S. Button[3], Elizabeth A. Hart[4],
Natalie M. Holmes[4], Kimberley Houghton[4],
Nina Badkar[2], Ellie Macey[2], Amy-Jayne Braggins[3],
Felicity C. Murray[2], Christopher D. Chambers[1] and
Rachel C. Adams[1]

[1]Brain Research Imaging Centre, Cardiff University, Cardiff CF24 4HQ, UK
[2]School of Psychology, University of Exeter, Exeter EX4 4QG, UK
[3]Department of Psychology, University of Bath, Bath BS2 7AY, UK
[4]School of Psychology, Cardiff University, Cardiff CF10 3AT, UK

  LT, 0000-0002-1463-9396; NSL, 0000-0003-1969-6637;
KSB, 0000-0003-4332-8789; CDC, 0000-0001-6058-4114;
RCA, 0000-0002-8053-0671

Inhibitory control training effects on behaviour (e.g. 'healthier' food choices) can be driven by changes in affective evaluations of trained stimuli, and theoretical models indicate that changes in action tendencies may be a complementary mechanism. In this preregistered study, we investigated the effects of food-specific go/no-go training on action tendencies, liking and impulsive choices in healthy participants. In the training task, energy-dense foods were assigned to one of three conditions: 100% inhibition (no-go), 0% inhibition (go) or 50% inhibition (control). Automatic action tendencies and liking were measured pre- and post-training for each condition. We found that training did not lead to changes in approach bias towards trained foods (go and no-go relative to control), but we warrant caution in interpreting this finding as there are important limitations to consider for the employed approach–avoidance task. There was only anecdotal evidence for an effect on food liking, but there was evidence for contingency learning during training, and participants were on average less likely to choose a no-go food compared to a control food after training. We discuss these findings from both a methodological and theoretical standpoint and propose that the mechanisms of action behind training effects be investigated further.

# 1. Introduction

The recent rise in overweight and obesity rates can primarily be attributed to the over-consumption of energy-dense foods that are high in fat, sugar and/or salt content [1]. One theoretical explanation for overeating has been provided by dual-process models, such as the reflective–impulsive model, which argues that behaviour is determined by the interaction of impulsive (*automatic*) and reflective (*controlled*) processes [2,3]. Specifically, over-consumption of energy-dense foods can be ascribed to heightened automatic biases for such foods, which can result in increased food intake if these automatic tendencies are not regulated via controlled processes [4]. For example, constant exposure to energy-dense foods in the environment can induce approach bias, or cravings, towards those foods and lead to over-consumption in individuals with limited self-control (trait or state), even when this behaviour is incompatible with health-related goals [5,6].

Theoretical frameworks, such as the reflective–impulsive model, have led to the development of behaviour change interventions for 'unhealthy' eating behaviours (e.g. overeating). These interventions are designed to target automatic and/or controlled processing, such as approach bias modification and inhibitory control training (ICT) [7,8]. One type of behaviour change interventions that has demonstrated recent success is food-specific ICT. Training has been shown to have therapeutic potential for reducing over-consumption of energy-dense foods and recent meta-analyses have cited small–medium effects when compared with control training [9,10]. Inhibitory control can broadly be defined as an individual's ability to inhibit impulses that are not compatible with long-term goals, such as losing weight [11]. The ability to inhibit such responses has been negatively associated with 'unhealthy' eating behaviours [12–15]. For example, Nederkoorn *et al.* [15] showed that strong implicit preferences for snacks paired with low 'inhibitory control capacity' predicted weight gain over 1 year.

ICT interventions have subsequently been used in an attempt to improve inhibitory ability and behaviourally related outcomes. Such interventions most commonly employ the go/no-go task [16,17] and stop–signal task [18,19], in which individuals are trained to inhibit motor responses when a cue is presented. In food-related ICT interventions, participants are instructed to make a speeded response to food stimuli (go/no-signal trial), but to withhold that response when a signal, or cue, is presented (stop/signal, trial). Stimulus–response mappings can be manipulated so that appetitive stimuli (e.g. energy-dense foods) are consistently paired with a stop signal, thus training an individual's ability to stop to these stimuli. These training tasks have been shown to reduce food consumption [20–24] and promote healthy food choices in the laboratory [25,26]. ICT protocols have even been associated with increased weight loss [27,28].

Although in both ICT paradigms, participants are required to inhibit responses towards target stimuli, the type of response inhibition that may be required is not necessarily the same. In go/no-go training, participants are presented with a cue/signal on 50% of the trials with zero-to-little delay, while in stop–signal training tasks, the signal is only shown on a minority of trials and its onset is delayed to maintain task difficulty (see [29] for comparison). Although go/no-go training may not necessarily tap into 'inhibitory control'[1] as a top-down process, meta-analyses have shown that compared to stop–signal training, it is more effective in changing eating-related behaviour [9,10].

There have been several suggested mechanisms of action behind ICT effects on behaviour, including the strengthening of food-specific inhibitory control and the reduction of motivational bias (incentive salience) for specific foods via a modulation of dopamine signalling in the brain's reward pathways [29,31,32]. It has also been observed that ICT can lead to changes in the hedonic value of foods, whereby appetitive foods are rated as less attractive and/or tasty after training (e.g. [25,27,33]). The devaluation of foods can in turn be explained by several, potentially interacting, mechanisms [29].

A prominent explanation was provided by the behaviour stimulus interaction (BSI) theory which posits that food stimuli are devalued when negative affect is induced to resolve the ongoing conflict between triggered approach reactions and the need to inhibit responses towards appetitive foods [33,34]. It was proposed that the *no-go devaluation effect* would occur only for positive/appetitive stimuli [33,34], but contradicting findings to this assumption have also been reported [35]. Studies have also provided evidence for a *go valuation effect*, whereby stimuli associated with go responses are evaluated more positively after training [33,36].

---

[1]We use the term 'inhibitory control training' throughout this manuscript to refer to both paradigms that are commonly reported as such in the literature, but we acknowledge that there is not enough evidence to suggest that go/no-go training taps into top-down inhibitory control mechanisms (e.g. automatic or controlled inhibition; see [30]).

Theoretically, the effects of ICT could also be explained by hard-wired neural connections between Pavlovian appetitive/aversive centres and go/no-go responses, respectively [37–39]. When a stimulus is consistently paired with a stop cue during training, a stimulus–stop association can be formed via associative learning [30]. This stop-associated stimulus can become devalued via a mutually facilitatory connection between a 'stop system' and the aversive system (increased avoidance). Similarly, the value of a stimulus consistently paired with go responses can be increased via the interaction between the 'go system' and the appetitive centre [40–42].

Explanatory accounts of ICT effects can, therefore, provide a theoretical ground for further investigation of automatic action tendencies towards appetitive stimuli after training. The BSI theory suggests that approach bias is reduced via devaluation during training to facilitate response inhibition. Similarly, in an associative stop system, go and stop responses/goals are directly linked to appetitive and aversive centres. Other accounts also support this expectation, as modulating the anticipated reward value of foods at a neural level could potentially lead to reduced approach bias for these foods [31]. Accordingly, one could assume that training does not only have an effect on the affective evaluation of trained foods, but also on the motivational, or reward, value of these foods and the individuals' tendencies to approach them for consumption.

Action tendencies are typically measured in an approach–avoidance task (AAT) by comparing individuals' reaction times to approach versus avoid a stimulus [43,44]. The AAT is assumed to capture *automatic* action tendencies when participants are instructed to respond to a task-irrelevant feature such as the orientation (portrait or landscape) of the presented picture, by pulling or pushing a joystick [45]. An approach bias can, therefore, be defined as the tendency to be faster at approaching a stimulus rather than avoiding it [45] and has been demonstrated for a variety of energy-dense foods in both obese and healthy-weight individuals [46–50]. Previous studies have used different methods for inferring food approach bias, such as the stimulus response compatibility (SRC) manikin task (e.g. [46,50]) and variants of the implicit association test (e.g. [47,48]), but there is some evidence to suggest that the AAT can capture food-related automatic action tendencies [51,52].

Although training has been shown to influence go/stop associations for food stimuli in terms of motor effort (e.g. [50–52]; see also [22]), to our knowledge, the effects of ICT interventions on approach bias for foods as measured by the AAT have not yet been investigated and this question has implications for both theoretical explanations of training outcomes and applied research in the context of eating behaviours. The current study attempts to answer this question by measuring the effects of go/no-go training on automatic action tendencies. A go/no-go task was used in which energy-dense foods were randomly assigned to one of three conditions. In the go condition, foods were consistently paired with a go response; in the no-go condition, foods were consistently paired with a no-go response; in the control condition, foods were inconsistently paired with go and no-go responses (50 : 50 mapping). Approach–avoid responses were recorded for each of these food conditions before and after training. Our primary hypothesis was that individuals would show reduced approach bias for no-go foods (H1a) and/or increased approach bias for go foods (H1b).

Consistent with previous literature, we included measures of impulsive food choice and food liking to investigate training effects on behaviour and food evaluation. It was expected that participants would show reduced choices for no-go foods (H2a) and increased choices for go foods, relative to control foods (H2b). It was also hypothesized that after training, the evaluations of no-go foods would be reduced compared to control foods (H3a—manipulation check), whereas the evaluations of go foods would be increased (H3b). As a manipulation check, we investigated contingency learning during training by exploring both error rates (H4a) and reaction times (H4b; e.g. [27]). All methods, confirmatory hypotheses and respective statistical tests (table 1) were preregistered (https://osf.io/wav8p/) as part of the GW4 Undergraduate Psychology Consortium which aims to promote collaborative and reproducible science for undergraduate students [53].

# 2. Material and methods

## 2.1. Participants

A total of 255 participants were recruited from the University campuses of Cardiff, Bath and Exeter via research participation schemes (e.g. Experimental Management system; EMS), advertisements and personal communication (see figure 2 for details). Participants recruited through participation schemes received course credits, whereas other individuals were offered entry into a prize draw for one of

**Table 1.** Preregistered hypotheses and respective Bayesian paired samples $t$-tests. $\Delta$AAT, change in approach–avoidance bias scores from pre- to post-training (ms) for go ($\Delta\text{AAT}_{\text{go}}$), no-go ($\Delta\text{AAT}_{\text{nogo}}$) and control foods ($\Delta\text{AAT}_{\text{control}}$); $p$, probability of choosing a go, no-go or control food; $\Delta$Liking, change in liking from pre- to post-training for go ($\Delta\text{Liking}_{\text{go}}$), no-go ($\Delta\text{Liking}_{\text{nogo}}$) and control foods ($\Delta\text{Liking}_{\text{control}}$); PCstop, proportion of correct responses on signal trials (i.e. stops) for no-go ($\text{PCstop}_{\text{nogo}}$) and control foods that appear on signal trials ($\text{PCstop}_{\text{control}}$), GoRT, correct go reaction time on no-signal trials for go ($\text{GoRT}_{\text{go}}$) and control foods presented on no-signal trials ($\text{GoRT}_{\text{control}}$).

| preregistered hypothesis | | directional $t$-test |
|---|---|---|
| H1. Training effects on automatic action tendencies | | |
| H1a | participants would show a reduction in approach bias for no-go foods compared to the control foods, from pre- to post-training | $\Delta\text{AAT}_{\text{nogo}} < \Delta\text{AAT}_{\text{control}}$ |
| H1b | participants would show increased approach bias for go foods compared to the control foods, from pre- to post-training | $\Delta\text{AAT}_{\text{go}} > \Delta\text{AAT}_{\text{control}}$ |
| H2. Training effects on impulsive food choices | | |
| H2a | the probability of choosing a no-go food would be lower than the probability of choosing a control food | $p(\text{no-go}) < p(\text{control})$ |
| H2b | the probability of choosing a go food would be higher than the probability of choosing a control food | $p(\text{go}) > p(\text{control})$ |
| H3. Training effects on food evaluations | | |
| H3a. | participants would show reduced liking for no-go foods relative to the control foods, from pre- to post-training | $\Delta\text{Liking}_{\text{nogo}} < \Delta\text{Liking}_{\text{control}}$ |
| H3b. | participants would show increased liking for go foods relative to the control foods, from pre- to post-training | $\Delta\text{Liking}_{\text{go}} > \Delta\text{Liking}_{\text{control}}$ |
| H4. Contingency learning during training | | |
| H4a. | the proportion of correct responses on signal trials would be greater for no-go foods compared to the control foods associated with a signal | $\text{PCstop}_{\text{nogo}} > \text{PCstop}_{\text{control}}$ |
| H4b. | go reaction times would be faster for go foods compared to the control foods presented on no-signal trials | $\text{GoRT}_{\text{go}} < \text{GoRT}_{\text{control}}$ |

three £20 shopping vouchers. A small proportion of participants who were recruited via personal communication completed the study outside laboratory settings (online). All participants were asked to refrain from eating for 3 h before the study. Participants had to be at least 18 years of age, be fluent in spoken and written English and have normal or corrected-to-normal vision, including normal colour vision. Participants were excluded if they were dieting at the time of the study, with a weight goal and timeframe in mind, if they had a current and/or past diagnosis of any eating disorder(s) or they had a body mass index (BMI) lower than 18.5 kg m$^{-2}$ (i.e. underweight category). The study was approved by the Ethics Committees of Cardiff University, University of Bath and University of Exeter.

## 2.2. Sampling plan

The required sample size was estimated based on a frequentist power analysis conducted for the primary outcome measure (i.e. change in approach–avoidance bias, from pre- to post-training, between go and no-go foods; H1a and H1b) and the stimulus devaluation manipulation check (i.e. change in food liking, from pre- to post-training, between go and no-go foods; H3). Both of these effect sizes were in the medium range and, therefore, calculations were based on the primary outcome measure (see electronic supplementary material, figure S1 for details). Based on *a priori* power calculations using G*Power [54], it was estimated that a total sample of 149 participants were necessary for 90% power ($\alpha = 0.005$, as recommended by Benjamin *et al.* [55]). Note that although the sampling plan was based on a conservative frequentist power analysis, the preregistered analyses followed a Bayesian framework and frequentist statistics are only reported in a supplementary manner. Bayes factor

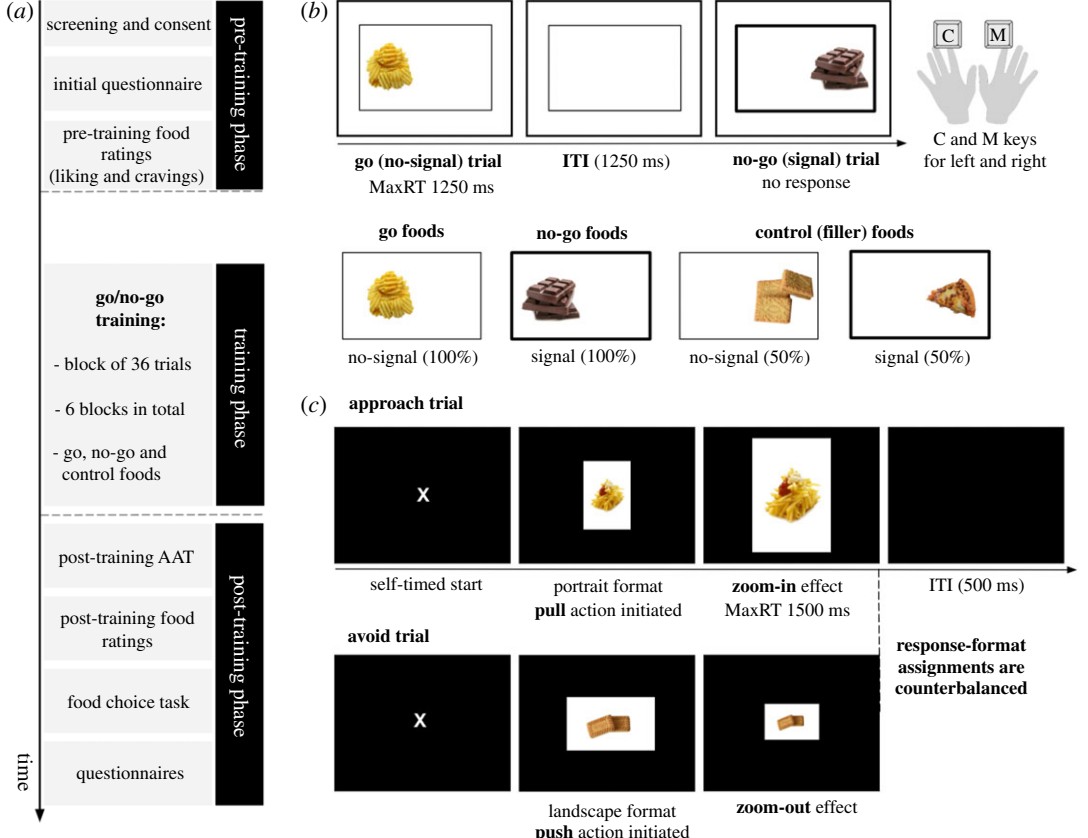

**Figure 1.** Schematic diagram of the study procedure, go/no-go training and approach–avoidance tasks. (*a*) After completing the screening and initial survey, participants rated all food stimuli (liking) and proceeded to perform the pre-training AAT blocks. In the training phase, participants completed six blocks of go/no-go training. The post-training AAT blocks were then presented and followed by food liking ratings. At the end of the study, participants completed a food choice task and several questionnaires, in random order. (*b*) The go/no-go training paradigm involved go (no-signal) and no-go (signal) trials that occurred with equal probability. On go trials, participants had to respond within 1250 ms by pressing the 'C' and 'M' keys to indicate the picture location (left or right, respectively). On no-go trials, participants were instructed not to respond at all. The inter-trial interval (ITI) was 1250 ms. Food categories were randomly assigned to three conditions. Go foods were only paired with no-signal trials and no-go foods were always associated with signal trials. Control foods were presented in both signal and no-signal trials (50 : 50). (*c*) In the AAT, participants were asked to respond according to the format of the presented picture (portrait or landscape). Response-format assignments were approximately counterbalanced across participants. As an example, on approach trials, a participant would have to pull the mouse towards them when the picture was in the portrait format (approach trial) and push it away from them when the picture was in the landscape format (avoid trial). Push and pull actions were paired with visual feedback, that is, zoom-out and zoom-in effects, respectively. The maximum reaction time (maxRT) was 1500 ms and the ITI was set to 500 ms. Participants clicked on a central 'X' to begin a trial (self-timed start).

analyses were favoured for drawing conclusions from the study, as they would allow us to interpret null outcomes as evidence of absence when traditional analyses would preclude such inferences.

## 2.3. Procedure

An overview of the study procedure is presented in figure 1*a*. After screening, eligible participants were provided with a short survey to record various sample characteristics (see *Survey and questionnaires*) and proceeded to rate all food categories on how much they like the taste (see *Food liking ratings*). Three blocks of the AAT (see *Approach–avoidance task*) were completed before the go/no-go training paradigm was performed (see *Go/no-go training*). Rated food categories were randomly assigned to three conditions for training: go, no-go and control, as shown in figure 1*b*. Post-training, participants were presented with another three blocks of the AAT, provided ratings for all food stimuli again and finally completed a short food choice task (see *Food choice task*). At the end of the study, several questionnaires were presented in random order and participants were debriefed about the aims of the

study. All study components were programmed using Inquisit Lab [56] and run online across data collection sites via Inquisit Web.

## 2.4. Go/no-go training

The Go/no-go training (GNG) paradigm involved go and no-go responses to six pre-selected energy-dense food categories. Food categories differed in terms of taste, so that three foods were savoury (i.e. pizza, crisps, chips) and three foods were sweet (i.e. biscuits, chocolate, cake). Two food categories were randomly assigned to each training condition (go, no-go, control foods) in the beginning of the experiment and food taste was counterbalanced so that each condition had one sweet and one savoury food. There were three training conditions according to the mapping of foods to signal (no-go) and no-signal (go) trials in the GNG. All go foods appeared in go (no-signal) trials and all no-go foods were presented in no-go (signal) trials (figure 1b). Control, or filler, foods appeared on both go and no-go trials with equal probability (50 : 50). Each food category had three exemplars which appeared twice in each block.

All foods were presented on either the left or right side of the screen within a rectangle for 1250 ms, which was the maximum reaction time (maxRT), as shown in figure 1b. Participants were asked to respond to the location of the food as quickly and as accurately as possible by pressing the 'C' and 'M' buttons on the keyboard with their left and right index fingers, respectively. The central rectangle remained on the screen throughout the training, including the inter-trial interval (ITI), which was 1250 ms. On signal trials, the rectangle turned bold, indicating that participants should withhold their response. This signal appeared on stimulus onset (i.e. no delay between stimulus and signal) and stayed on the screen until the end of the trial. A correct response on go (no-signal) trials was registered when participants responded accurately to the location of the food within the maxRT window and a successful stop (i.e. correct no-go trial) was considered when participants did not respond at all. Incorrect responses on go trials refer to either a wrong location judgement or a missed response. Left and right responses were counterbalanced across all manipulated variables for each type of trial. Training was split into six blocks of 36 trials (216 trials in total) and lasted approximately 10 min with inter-block breaks (15 s). Task practice included 12 trials of go and no-go responses (50 : 50) and participants responded to the location of grey squares, instead of food pictures. For the practice trials, accuracy feedback was provided during the ITI.

## 2.5. Approach–avoidance task

The AAT was adapted from an existent paradigm [43,44], which involves 'pull' (i.e. towards self) and 'push' (i.e. away from self) movements of a joystick. Each type of motor response is paired with visual feedback so that when the joystick is pulled, the image gets bigger (zoom-in) and when it is pushed, the image gets smaller (zoom-out). This 'zooming' feature acts as an exteroceptive cue of either an approach or avoidance response [57] and complements the proprioceptive properties of the task, where responses requiring arm flexion and extension correspond to approach and avoidance trials, respectively [44]. The evaluation-irrelevant feature of the paradigm was also incorporated, and participants responded according to the format of the picture [52].

For the current task, AAT responses involved 'push' and 'pull' movements of the computer mouse (adaptation of the joystick version). Food stimuli were presented in the centre of the screen and participants were instructed to pull the mouse towards them or push the mouse away from them according to whether the image was in portrait or landscape format (figure 1c). Response-format assignments were approximately counterbalanced across participants (45.4% portrait-approach, 54.6% landscape-approach). Instructions highlighted moving the mouse cursor until it reaches the end of the screen (top or bottom edge) for a correct response to be registered and making smooth whole-arm movements. Participants had 1500 ms to respond after the stimulus appeared. Each trial started with a central 'X' on the screen and participants had to click on it to begin (self-timed start). The ITI was 500 ms and there was no delay between the 'X' click response and the stimulus onset.

Each AAT block consisted of 72 trials and go, no-go and control foods appeared with equal probability for both 'pull' (approach) and 'push' (avoid) responses. There were 12 approach and 12 avoid trials for each training condition (e.g. no-go) and within those trials, there were six savoury and six sweet foods presented (i.e. three exemplars repeated twice). Three AAT blocks were performed before training (AAT$_{pre}$) and three after training (AAT$_{post}$). Two constraints were placed on the quasi-random order of the trials within an AAT block (cf. [42]). There were no more than three images of the same food category being presented consecutively and no more than three trials with the same

picture format in sequence. AAT practice consisted of 10 trials with grey rectangles instead of food stimuli and accuracy feedback. The screen background for the AAT was black and the task lasted approximately 15 min, including the inter-block 15 s breaks, where participants received a reminder of the main instructions.

## 2.6. Food liking ratings

Participants provided food liking ratings (cf. [27]) before and after training using a visual analogue scale (VAS). They rated all foods included in the GNG paradigm according to how much they liked the taste (How much do you like the taste of this food?), ranging from 0 (not at all) to 100 (very much). Task instructions encouraged participants to imagine they were tasting the food in their mouth and then rate how much they liked the taste [58]. The cursor position was initially set to 50 for each food. The order of the presented foods was randomized and each block consisted of 18 trials.

## 2.7. Food choice task

Impulsive food choices were assessed using a food choice task adapted from Veling *et al.* [25], which included all food categories from the GNG paradigm (two exemplars per category). The 12 foods were presented on a grid layout, in a random order, and participants had 10 s to select three foods that they would like to consume the most at that specific time, by clicking on them with the computer mouse. Participants were asked to click on a 'start' button to begin the trial and when a response was registered, the selected food stimulus disappeared from the screen. This task element was introduced to prevent participants from deliberating on their choices and changing their initial responses, which could mean that *impulsive* food choices were no longer measured. Task instructions did not mention whether the nature of their choices would be consequential or hypothetical (i.e. whether they would get a food item at the end of the study or not).

## 2.8. Survey and questionnaires

Eligible participants were presented with an initial survey to record demographics and other variables for exploratory analyses. The survey consisted of self-reported height and weight measurements to calculate a participant's BMI ($kg\,m^{-2}$), gender, the number of hours since their last meal ('less than 3 h ago', '3–5 h ago', '5–10 h ago', 'more than 10 h ago') and hunger at the time of the study (Likert; 1 = 'Not at all' to 9 = 'Very'). Several questionnaires were completed by the participants at the end of the study for exploratory analyses, as part of the undergraduate student projects of the GW4 Undergraduate Psychology Consortium 2017/2018 (see electronic supplementary material, figure S2).

# 3. Analyses

## 3.1. Measures and indices

Performance in the AAT blocks was considered only for correct responses. The median RTs for 'push' and 'pull' responses from all training condition levels were calculated for each participant. Medians were used instead of means as they are less sensitive to outliers in RT distributions (also see [44,59]). The approach–avoidance bias score for each condition was calculated as the difference between the median completion RTs (see electronic supplementary material, figure S3) for 'push' and' pull' responses (MedianRT$_{push}$–MedianRT$_{pull}$). Bias scores were computed for both AAT$_{pre}$ and AAT$_{post}$ blocks. Positive scores indicate an approach bias towards the foods of interest and negative scores reflect avoidance for those foods. Change scores for approach–avoid biases from pre- to post-training (ΔAAT) were calculated for preregistered analyses (H1—table 1).

 In the food choice task, participants were required to choose three foods out of 12 and selections could vary in their number for each training condition (go, no-go, control). Food choices were, therefore, normalized according to the total number of responses per participant (i.e. proportion) and compared across training conditions (H2). Food liking scores were averaged (mean) across the two foods per training condition (i.e. sweet and savoury foods for go, no-go and control conditions) and the three exemplars of each food. Changes in food liking from pre- to post-training (ΔLiking) were compared for preregistered analyses (H3). Pre-training scores were subtracted from post-training

scores and negative values represent a reduction in liking. All preregistered data exclusions can be found in the electronic supplementary material, figure S4.

## 3.2. Preregistered analyses

Data pre-processing and analyses were conducted in R [60] via RStudio [61] and JASP [62]. Preregistered analyses are described under their pre-specified hypotheses in table 1. For all Bayesian paired samples *t*-tests mentioned hereinafter, a prior with the $\sqrt{2}/2$ scale parameter for the half-Cauchy distribution was used. The evidential value of confirmatory findings was solely determined by the Bayesian tests outlined in this section and we followed the guidelines by Lee & Wagenmakers [63] for BF grades of evidence ($BF_{10} \geq 3$ would indicate moderate evidence for the alternative hypothesis, while $BF_{10} \leq 1/3$ would indicate moderate evidence for the null). Frequentist tests were conducted in order to further the reproducibility of findings (e.g. potential use in meta-analyses). Paired samples *t*-tests were two-tailed,[2] in line with the reported power analysis. Note that supplementary analyses were conducted for our preregistered tests regarding training effects (H1–H3) with an adjusted prior distribution that was more compatible with the expected effect size of interest ($d_z = 0.34$; see electronic supplementary material, figure S1 for details). The results from these supplementary analyses were overall consistent with our preregistered analyses (see electronic supplementary material, figure S6).

## 3.3. Deviations from preregistration

We report two minor, non-consequential, deviations to the preregistered analysis plan concerning repeated-measures ANOVAs for H1 and H3 and associated Bonferroni corrections for frequentist *t*-tests following the ANOVAs. The evidence for the distinct hypotheses about the data would be provided by two planned directional *t*-tests (H1a, H1b and H3a, H3b) and the ANOVAs were not required. The approach of conducting ANOVAs and 'post hoc' comparisons with the Bonferroni corrections was only added for frequentist analyses as part of the students' dissertations. These deviations do not affect the reported results or their interpretation.

# 4. Results

## 4.1. Sample characteristics

After exclusions (figure 2), the final sample consisted of 163 participants[3] (80.98% female). Participants' average BMI was in the 'healthy-weight' range ($M = 22.88$, s.d. = 2.98, range = 18.54–32.36) and their mean age was 22.39 (s.d. = 9.04, range = 18–59). A total of 108 participants (66.26%) reported that they had their last meal 3–5 h before the study, while 24 participants (14.72%) did not adhere to the instruction not to eat 3 h before. As expected, hunger levels were not particularly high ($M = 5.70$, s.d. = 2.22). Participant's scores on several questionnaires can be found in electronic supplementary material, table S1.

## 4.2. Findings from confirmatory analyses

### 4.2.1. Training effects on automatic action tendencies

Approach–avoidance bias scores pre- and post-training across conditions have been visualized using rainclouds [64,65] in figure 3 and all results for paired comparisons are shown in table 2. There was *moderate* evidence that the change in bias scores for no-go foods ($\Delta AAT_{nogo}$; $M = -3.31$, s.d. = 62.91) was not reduced compared to the change for control foods ($\Delta AAT_{control}$; $M = -1.81$, s.d. = 59.55). Similar to H1a, there was *strong* evidence for the null compared to the alternative for H1b. The change in bias scores for go foods ($\Delta AAT_{go}$; $M = -10.47$, s.d. = 59.57) was not greater than the change for control foods.

---

[2]Contingency plans were not considered in case the normality assumption was violated for paired *t*-tests (Shapiro–Wilk test: $p \leq 0.005$), but appropriate exploratory analyses were conducted and reported in the *Results* section.

[3]As part of our data collection procedure, the data were inspected for exclusions after several group testing sessions ($n > 100$) in order to check the number of participants, we would need to recruit to meet our target sample size ($n = 149$). For the majority of the sessions, we had an unexpected rate of exclusions, but towards the end of data collection, the number of participants that were not excluded due to low accuracy in the AAT was greater than anticipated. Due to the nature of the data collection setting (i.e. group slots), this resulted in the final sample size exceeding the initial target by 14 participants.

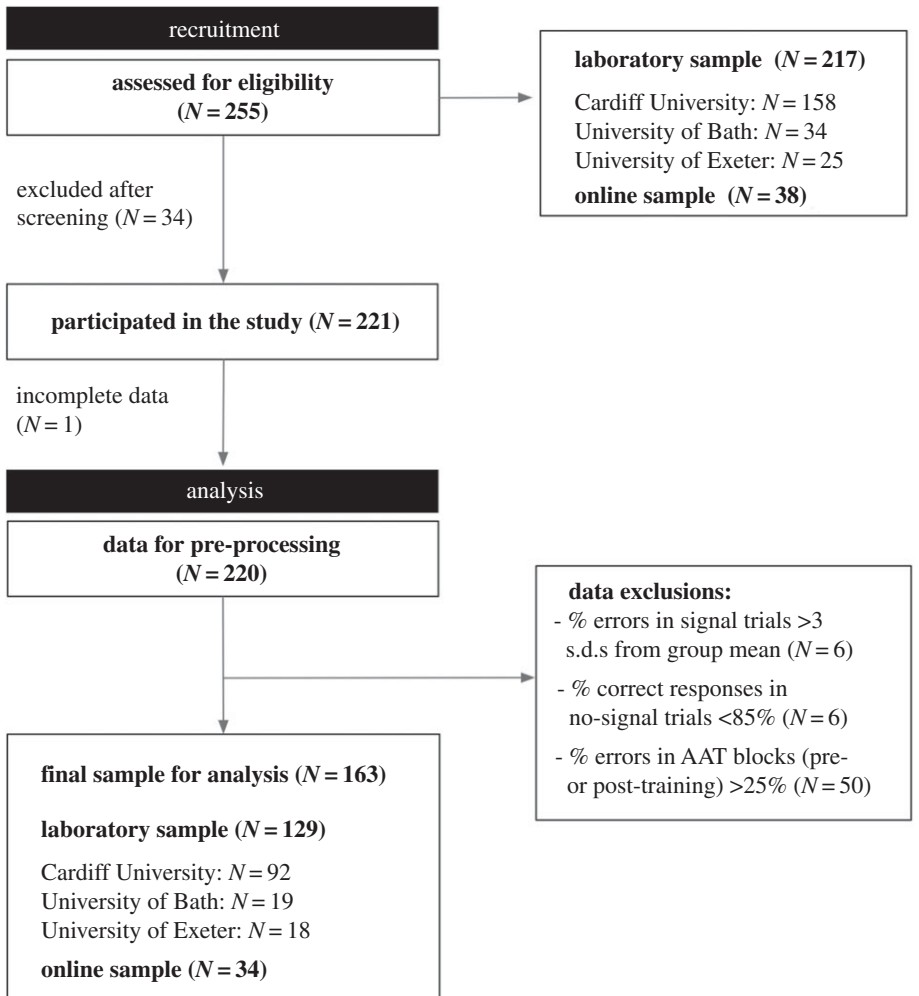

**Figure 2.** Flow diagram of recruitment and participant-level data exclusions. There were 255 individuals recruited and assessed for eligibility across laboratory sites and online via personal communication. Thirty-four participants were excluded after screening for not meeting the advertised inclusion/exclusion criteria and datasets were obtained from 221 participants. The online sample was recruited by the University of Bath and University of Exeter. One participant was excluded for providing incomplete data and 220 datasets were submitted for pre-processing and inspection. First, we examined performance in the go/no-go training task. There were no participants with a mean reaction time on no-signal trials (GoRT) greater than 3 s.d. from the group mean and there were no cases of consistently missed (i.e. default option of 50) responses on food rating trials. Six participants had a percentage of errors in signal trials that was greater than 3 s.d. from the group mean and six participants also had a percentage of correct responses in no-signal trials lower than 85%. Note that some participants met more than one exclusion criterion. Performance in the AAT was inspected and 50 participants were excluded as their percentage of errors in either the pre- or post-training blocks was greater than 25%. The final sample consisted of 163 participants.

### 4.2.2. Training effects on impulsive food choices

The effect of training on impulsive food choices was examined for no-go and go foods compared to control foods, as stated in H2a and H2b, respectively. One participant did not complete the food choice task. There was *extreme* evidence that the probability of choosing a no-go food ($M = 0.21$, s.d. = 0.27) was lower than the probability of choosing a control food ($M = 0.36$, s.d. = 0.31) after training (table 2). There was only *anecdotal* evidence that the probability of choosing a go food ($M = 0.44$, s.d. = 0.33) was not higher than the probability of choosing a control food.

### 4.2.3. Training effects on food liking

As a first manipulation check and secondary training outcome, it was investigated whether GNG changed the evaluations of no-go foods during training compared to the evaluations of control foods (figure 4). The change in liking scores from pre- to post-training for no-go foods ($\Delta\text{Liking}_{nogo}$; $M = -4.16$; s.d. = 9.51) was

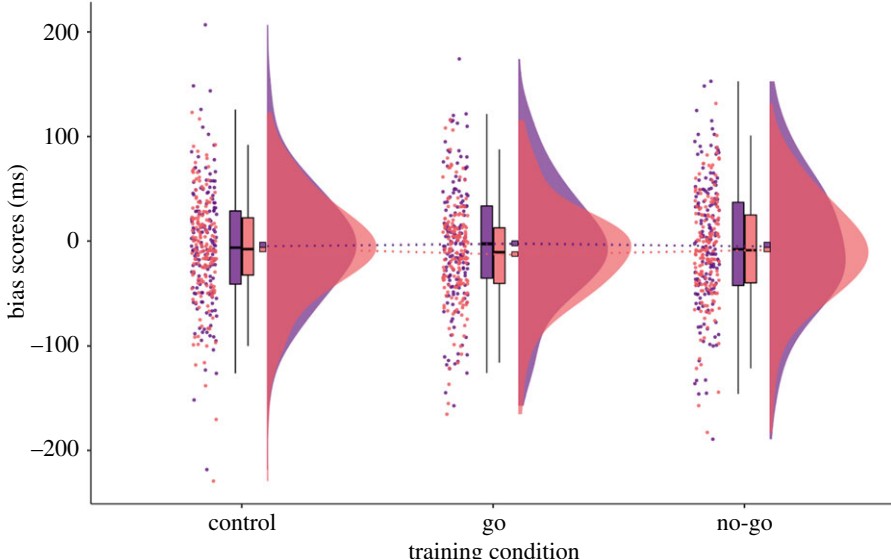

**Figure 3.** Raincloud plot of the approach–avoidance bias scores pre- and post-training across training conditions. There were no differences between the sample mean changes in approach–avoidance bias scores for no-go and go foods compared to control foods, as shown by the dotted lines. At a closer inspection, individual bias scores do not seem to be clustered around the positive end of the distribution as it would be expected for appetitive energy-dense foods, but actually show less dispersion around zero. Exploratory analyses confirmed that baseline bias scores for go, no-go and control foods did not statistically deviate from zero (see *Baseline approach bias scores*). The 'split-half violin' elements in the plot show smoothed distributions and boxplot vertical lines represent the range, excluding outliers based on IQR. Square boxes have been added to depict the sample means, connected with dotted lines across training conditions.

slightly reduced compared to change in liking for control foods ($\Delta Liking_{control}$; $M = -2.61$, s.d. $= 8.77$), and there was only *anecdotal* evidence for this effect (H3a; table 2). The change in liking scores from pre- to post-training for go foods ($\Delta Liking_{go}$; $M = -2.87$, s.d. $= 10.15$) was not greater than the change for control foods. Instead, there was *strong* evidence for the null hypothesis compared to the alternative (H3b).

### 4.2.4. Contingency learning during training

In order to validate whether the implemented GNG paradigm led to stimulus–response associations (i.e. contingency learning manipulation check), we tested whether the percentage of correct responses for no-go foods (i.e. successful inhibitions) would be greater compared to the percentage of correct responses for control foods associated with signal trials (H4a). There was *extreme* evidence that participants had on average a higher proportion of successful inhibitions for no-go foods ($PCstop_{nogo}$; $M = 0.97$, s.d. $= 0.03$) than control foods ($PCstop_{control}$; $M = 0.96$, s.d. $= 0.04$). For H4b, it was examined whether mean reaction times would be reduced for go foods ($GoRT_{go}$; $M = 507.00$, s.d. $= 70.48$) compared to control foods associated with no-signal trials ($GoRT_{control}$; $M = 515.00$, s.d. $= 75.51$) and there was *extreme* evidence for such an effect. Therefore, contingency learning was observed in the employed GNG paradigm for both reaction time and accuracy outcomes.

## 4.3. Findings from exploratory analyses

### 4.3.1. Baseline approach bias scores

Performance in the AAT was inspected further to check if an approach bias for foods was present and whether error rates differed across conditions at baseline (see electronic supplementary material, figure S5 for AAT accuracy analysis). Although the sample means for AAT bias scores were negative for all foods (go foods: $M = -2.32$, s.d. $= 58.14$; no-go foods: $M = -4.75$, s.d. $= 60.58$; control foods: $M = -4.48$, s.d. $= 52.25$), individual data points show less dispersion close to zero (figure 3), suggesting that, on average, participants did not have any approach bias towards the foods, as measured by the AAT in this study.

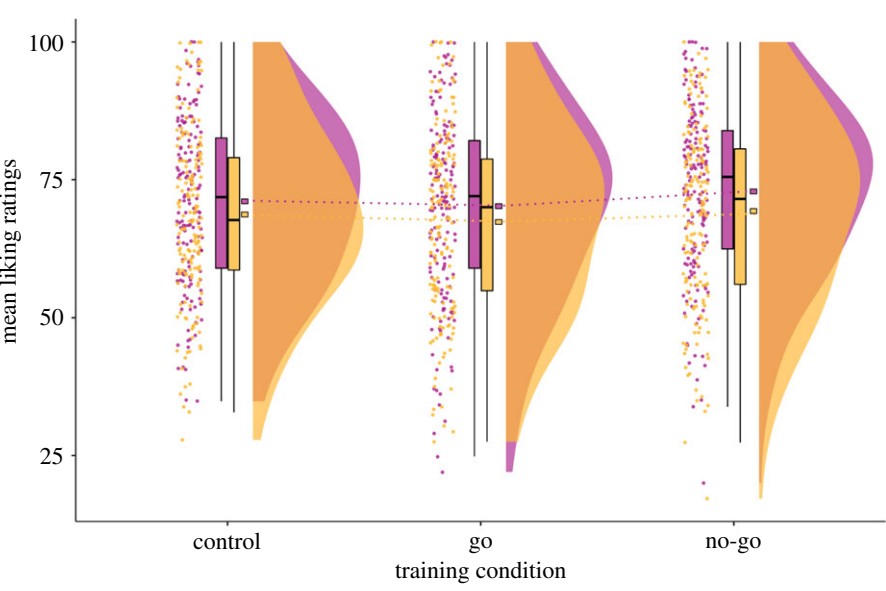

**Figure 4.** Raincloud plot of the mean liking ratings pre- and post-training across training conditions. Liking ratings were registered on a VAS ranging from 0 to 100. There appears to be a general trend of devaluation across training conditions which could be attributed to regression to the mean (but see *Discussion*), although the plot shows that on average, participants did not like the taste of the foods 'very much' (e.g. greater than 80). Given that the foods were not selected based on participants' ratings and we used a fixed set of stimuli, we consider that they were adequately appetitive. The 'split-half violin' elements in the plot show smoothed and trimmed distributions and boxplot vertical lines represent the range, excluding outliers based on the IQR. Square boxes have been added to depict the sample means, connected with dotted lines across training conditions.

**Table 2.** Results for all confirmatory hypotheses and respective statistical tests. Evidence is interpreted for the alternative hypothesis (H1) compared to the null (H0) and vice versa (see §4.2 for guidelines). The effect size is represented by Cohen's *d*. ΔAAT, change in approach–avoidance bias scores from pre- to post-training (ms) for go, no-go and control foods; *p*, probability of choosing a go, no-go or control food; ΔLiking, change in liking from pre- to post-training for go, no-go and control foods; PCstop, proportion of correct responses on signal trials (i.e. stops) for no-go and control foods that appear on signal trials; GoRT, correct go reaction time on no-signal trials for go and control foods presented on no-signal trials.

| hypothesis | BF$_{10}$ | *t* | d.f. | *p*-value | *d* | 95% CI for *d* | interpretation |
|---|---|---|---|---|---|---|---|
| H1a. $\Delta AAT_{nogo} < \Delta AAT_{control}$ | 0.11 | −0.25 | 162 | 0.805 | −0.02 | [−0.17, 0.13] | *moderate* evidence for H0 |
| H1b. $\Delta AAT_{go} > \Delta AAT_{control}$ | 0.04 | −1.35 | 162 | 0.179 | −0.11 | [−0.26, 0.05] | *strong* evidence for H0 |
| H2a. $p(\text{no-go}) < p(\text{control})$ | 247.78 | −3.93 | 161 | <0.001 | −0.31 | [−0.47, −0.15] | *extreme* evidence for H1 |
| H2b. $p(\text{go}) > p(\text{control})$ | 0.85 | 1.82 | 161 | 0.070 | 0.14 | [−0.01, 0.30] | *anecdotal* evidence for H0 |
| H3a. $\Delta Liking_{nogo} < \Delta Liking_{control}$ | 2.65 | −2.38 | 162 | 0.019 | −0.19 | [−0.34, −0.03] | *anecdotal* evidence for H1 |
| H3b. $\Delta Liking_{go} > \Delta Liking_{control}$ | 0.07 | −0.37 | 162 | 0.715 | −0.03 | [−0.18, 0.13] | *strong* evidence for H0 |
| H4a. $PCstop_{nogo} > PCstop_{control}$ | 140.25 | 3.77 | 162 | <0.001 | 0.30 | [0.14, 0.45] | *extreme* evidence for H1 |
| H4b. $GoRT_{go} < GoRT_{control}$ | 3973.21 | −4.66 | 162 | <0.001 | −0.37 | [−0.52, −0.21] | *extreme* evidence for H1 |

This assumption was tested by examining whether baseline bias scores statistically deviated from zero using Bayesian one-sample $t$-tests with the default prior settings for the two-sided alternative hypothesis that the population mean was not equal to the test value ($\neq 0$). We found *moderate* evidence that participants' bias scores (across all foods; $M = -2.77$, s.d. $= 44.71$) at baseline did not deviate from zero [$BF_{01} = 8.43$; $t(162) = -0.79$, $p = 0.430$, $d = -0.06$, 95% CI for $d = -0.22$, 0.09].

As bias scores calculated from *completion* times may be 'contaminated' by motor demands in this version of the AAT, which requires computer mouse movements and arm flexion/extension, it is possible that *initiation* times may be more sensitive to capturing automatic action tendencies. Movement initiation was registered when participants had moved their mouse cursor since starting a trial (see electronic supplementary material, figure S3 for details). Therefore, the one-sample $t$-test was also conducted for baseline bias scores calculated using median initiation times, instead of median completion times. Consistent with the results presented above, there was *moderate* evidence that baseline bias scores ($M = -2.33$, s.d. $= 35.56$) did not deviate from zero [$BF_{01} = 8.13$; $t(162) = -0.84$, $p = 0.404$, $d = -0.07$, 95% CI for $d = -0.22$, 0.09].

### 4.3.2. Reliability of bias scores

Evidence for the absence of baseline approach bias as measured by the AAT could indicate that the internal reliability of the measure is questionable. The split-half method was used to quantify the internal reliability, or consistency, of the AAT bias scores (MedianRT$_{push}$−MedianRT$_{pull}$) at baseline. We used a permutation-based split-half approach and computed split-half estimates for 10 000 random splits of the correct completion RTs at baseline [66]. The Spearman–Brown-corrected reliability estimate ($\rho_{SP}$) was 0.67, 95% CI [0.58, 0.75].

Although completion RTs were used for the preregistered analyses, we also ran this analysis for AAT initiation times to test whether questionable internal consistency was associated with potential motor demands of completing whole-arm mouse movements. The Spearman–Brown-corrected estimate for AAT bias scores at baseline based on initiation RTs was in the same range; $\rho_{SP} = 0.69$, 95% CI [0.61, 0.77]. The reliability estimates for bias scores calculated from both completion times and initiation times were not very high (e.g. $\geq 0.8$) and we, therefore, acknowledge that the internal reliability of the AAT bias scores[4] as an outcome measure in ICT studies should be investigated further.

## 5. Discussion

The primary aim of the study was to investigate whether go/no-go training (GNG) can have an effect on automatic action tendencies. This research question was based on previous theoretical ground which links approach bias to response inhibition during ICT tasks, such as the BSI theory and the associative stop system [29,33,34,38,39]. In line with previous literature, we also measured participants' food evaluations and impulsive choices as secondary outcomes and examined contingency learning during training as a manipulation check.

In this study, we show that the go/no-go training protocol with pre-selected energy-dense foods was partially effective and its effects on automatic action tendencies require further investigation. Specifically, contingency learning during training occurred as expected and participants were less likely to choose foods that were consistently paired with no-go responses compared to control foods, but liking and approach bias for these foods were not reduced. Exploratory analyses indicated that participants did not have any approach bias for the foods at baseline, as measured by the AAT, and that the reliability of the calculated bias scores warrants caution in interpreting the null findings regarding the effects of training on automatic action tendencies. All findings and their potential explanations are discussed together with acknowledged limitations of the study design and directions for future research.

As a primary outcome measure, the change in bias scores from pre- to post-training was examined across training conditions using an AAT. The results from the preregistered analyses showed that GNG did not have an effect on automatic action tendencies; there was *moderate* evidence that approach bias for no-go foods was not reduced relative to control foods after training (H1a) as well as *strong* evidence that approach bias for go foods was not increased compared to control foods after training (H1b). Although such effects may not have been previously investigated, there is empirical

---

[4]Here, we only refer to the calculated bias scores (*measurement*) and not the AAT as an indirect *measure* of automatic action tendencies [67].

evidence to suggest that the AAT may be sensitive enough to capture AAT training-induced changes in action tendencies[5] [69,70]. In order to fully understand these 'null' findings, we must first take a closer look at potential training mechanisms and positive outcomes.

First, the manipulation check for contingency learning during training was successful which indicates that stimulus–response associations were formed during the GNG paradigm consistent with previous literature (see [27,60]). There was *extreme* evidence that GoRTs on correct no-signal trials were reduced for go foods compared to control foods and that the percentage of correct responses on signal trials were greater for no-go foods relative to control foods. Further evidence for the efficacy of training stemmed from our secondary outcome measure. Impulsive food choices were assessed via an adapted food choice task [25] after training and we found *extreme* evidence that the probability of choosing a no-go food was lower than the probability of choosing a control food (H2a). Meanwhile, there was *anecdotal* evidence that the probability of choosing a go food was not higher relative to the probability of choosing a control food (H2b). Studies that have found increased food choices for go food items employed a different paradigm, namely cued-approach training [26,62] and cannot directly contrast the finding reported here. From an applied perspective, it should be noted that the training protocol in this study included only energy-dense foods and future studies could pair healthy foods with go responses to test whether (and how) training can promote healthier food choices in the laboratory [26,71].

Reduced choices for no-go foods are consistent with previous studies that have used both go/no-go and stop–signal task paradigms [25,71–73]. For example, Chen *et al.* [71] showed that following a single training session, participants were more likely to choose go foods than no-go foods up to one week later. Importantly, previous research has indicated that training effects on food choice may be mediated by the devaluation of no-go foods [25], although in this study, there was not conclusive evidence for a no-go devaluation effect as discussed further below (also see review by Veling *et al.* [29]).

Recent evidence suggests that training effects may only be reliable for speeded, and not for deliberate, food choice [26,71], which indicates that demand characteristics would not affect the results in this study even if a proportion of participants was aware of stimulus–response contingencies after training (i.e. cake was a no-go food so I will not choose it). Previous research has further shown that memory of stimulus–response contingencies did not affect food choice outcomes [26]. Although the food choice task in this study required participants to respond within a time limit, future replications and/or extensions of these findings could still employ other impulsive choice measures that are less prone to strategic responding, as, for example, the speeded binary food choice task which involves multiple choice combinations and stricter time windows (see [74]).

In line with previous studies where go/no-go training led to robust food devaluation effects (e.g. [23,25,31]), it was expected that the change in the mean tastiness ratings (i.e. food liking) for no-go foods from pre- to post-training would be reduced compared to the change in ratings for control foods. Preregistered analyses showed only *anecdotal* evidence that no-go foods were rated less positively after training compared to control foods (H3a). Similarly, participants did not show increased liking for go foods relative to control foods, from pre- to post-training (H3b). The visualization of liking data hinted at a general devaluation trend for all foods (figure 4), which has also been observed in previous studies (e.g. [27,33,75]). This could be attributed to the regression to the mean, but we advocate that the potential of over-exposure effects in ICT protocols that repeatedly present a limited number of energy-dense foods should be formally addressed in future research.

A potential explanation for the inconclusive evidence regarding stimulus devaluation effects is that we used a fixed set of energy-dense foods (e.g. pizza, cake, crisps) that may not have been *highly* appetitive for many participants[6] [33]. Although this is a key assumption of the BSI theory, recent findings imply that no-go devaluation effects can be observed for low-rated food items [35]. An important design parameter that we should consider for the devaluation effects in applied ICT studies is the 'meaningfulness' of the stimulus–response pairings, as, for example, when there is consistent pairing of healthy and unhealthy foods with go and no-go responses, respectively [27,75,76]. In this study, participants could not attach meaning to the stimulus–response pairings (e.g. approaching an 'unhealthy' food is not desirable) as energy-dense foods were presented across all training conditions. Therefore, as already recommended, future studies could benefit from introducing consistent pairings of healthy foods with go responses in similar paradigms. It may also be worth investigating whether

---

[5]A significant change in approach–avoidance bias scores was not observed in another series of experiments [68].

[6]Despite the lack of a tailored food selection process (e.g. [33]) the three food sets (control, no-go, go) were approximately matched in terms of average liking before training, as shown in figure 4.

actual food liking (rating the flavour of foods in the laboratory) would be a more critical determinant of training effects as opposed to ratings of *expected* liking/taste.

Notably, a critical limitation of the present design may be the addition of an AAT before the post-training liking ratings because it can potentially interfere with the GNG manipulations. As task order was not counterbalanced across participants, we could assume that the AAT may have counteracted the consistent stimulus–response mappings of the GNG. For example, approaching no-go foods during the AAT could result in no-go foods not being associated with 100% inhibition and this could in turn affect subsequent explicit food evaluations. This warrants further caution for including behavioural tasks in ICT protocols that tap into potentially similar motor responses and/or mechanisms (see also [77]).

Although this study provides preliminary evidence that training did not influence automatic action tendencies, there were several findings from exploratory analyses regarding the AAT bias scores that may explain the absence of the expected effect and yield methodological considerations for future studies. We found that overall baseline bias scores did not statistically deviate from zero, which suggests that either participants in this sample (averagely healthy BMI) did not have any approach bias for the selected foods in the first place or that the employed variant of the AAT was not sensitive enough to capture both baseline bias and potential effects of training. An exploratory analysis indicated that the internal (split-half) reliability of the AAT bias scores at baseline may be questionable in this context ($0.6 \leq \rho_{SP} < 0.7$) and should indeed be investigated further. We should also mention that a total of 50 participants were excluded due to high error rates in the AAT (greater than 25%).

To explore this issue further, we considered that the response modality for the AAT in this study could have affected the reliability of the bias scores. We found a recent study that employed an AAT with the same response modality (i.e. computer mouse instead of joystick) and provided measures of bias for smoking-related stimuli before and after training [78]. In this study, there was evidence for neither approach bias at baseline nor a change in approach bias after training. The authors reported 'inadequate' split-half reliabilities for AAT scores for both smoking-related and smoking-unrelated stimuli, and when applying the Spearman–Brown correction, their estimates were very similar to the ones reported here, as, for example, an estimate of 0.63 for smoking-related stimuli. Consistent with our observations about AAT performance, the authors also reported relatively high error rates compared to other studies (e.g. 11% at baseline) and indicated that the use of the computer mouse could have increased measurement error [78]. More research is required to examine the conditions under which performance in this AAT variant can be improved in terms of accuracy and arm movements.

Another essential consideration for AATs is the variability in methodology in the literature which suggests that different task parameters may have contributed to the present findings (see Phaf *et al.* [79] for meta-analysis). For example, Lender *et al.* [80] found that the irrelevant feature of the AAT did not lead to robust approach bias, compared to relevant-feature variants which require participants to pay attention to the content of the stimuli. It is, therefore, possible that baseline bias in this study was not captured because many participants were simply categorizing the format of the picture (portrait or landscape) and did not pay attention to the content of the picture (i.e. specific foods). The use of the implicit instructions in the AAT could then affect the reliability of the bias scores (also see [81]).

We, therefore, propose that this research question is worth pursuing further in future empirical studies that address the above methodological considerations. The AAT could be tailored with explicit instructions that involve judgements on specific food categories or food/non-food stimuli and both the GNG and AAT tasks could include stimuli that are matched in liking and selected by the participants. Alternatively, future studies could employ other measures of motivational bias, such as the relevant stimulus–response compatibility (R-SRC; e.g. see Field *et al.* [82]). Regarding the specific GNG protocol in this study, future studies could use tasks that have been shown to produce robust no-go devaluation effects, such as the paradigm reported in Chen *et al.* [33].

# 6. Conclusion

Here, we present the first study to empirically investigate the theoretically proposed effects of go/no-go training on action tendencies. Training had an effect on participant's impulsive choices as the probability of choosing a no-go food was lower than the probability of choosing a control food after training, and there was conclusive evidence for contingency learning during training. However, there was only *anecdotal* evidence for a no-go devaluation effect and evidence for the absence of training effects on participants' automatic action tendencies towards trained foods (go and no-go relative to control).

We believe that the null findings presented here can shed light onto methodological and theoretical issues that need to be explored further. From a theoretical standpoint, there could be a link between stimulus devaluation during ICT training and automatic action tendencies. If a tendency to approach an appetitive food is reduced during go/no-go training in order for response inhibition to be successful, the approach bias towards food stimuli associated with signal trials could be indirectly affected by this process. Nevertheless, there are potential methodological limitations regarding the design of the AAT as an indirect measure of motivational bias in this context that need to be addressed before drawing any conclusions. The issue of operationalization in this research area may require more empirical attention as it would be of theoretical and applied interest to know whether go responses and approach motivation towards foods in the GNG (e.g. see BSI theory; [34]) can be directly mapped onto automatic action tendencies, as measured by other experimental tasks in the literature.

On a final note, it is worth mentioning that there are various methodological parameters and protocols that can be implemented for both ICT and measurement of approach–avoidance bias and this can pose an important replicability issue. It is recommended that novel findings, irrespective of statistical significance, are replicated and/or extended in a rigorous and reproducible manner, in an effort to reduce selective reporting and publication bias in this line of research [83,84]. Similarly, future research could further explore the role of individual differences in training outcomes and complementary or underlying mechanisms (approach bias, stimulus devaluation) by measuring factors such as restrained and external eating (e.g. [21,23,46,85]).

Ethics. This project was led by researchers at Cardiff University and received ethical approval by the Ethics Committee at the School of Psychology (EC.17.10.10.4995GR). For student participation, the study was further approved by the Ethics Committees at the University of Bath (17-254GR) and the University of Exeter (eCLESPsy000276). Informed consent was obtained from all participants.

Data accessibility. All study data and analysis scripts are freely available on the Open Science Framework (https://osf.io/hz2nb/). The study protocol was preregistered prior to data collection at https://osf.io/wav8p.

The data are provided in electronic supplementary material [86].

Authors' contributions. L.T. and R.C.A. designed the research, analysed and interpreted the data and drafted and finalized the manuscript. E.A.H., N.M.H., K.H., N.B., E.M., A.-J.B. and F.C.M. made substantial contributions to the study design, acquisition and interpretation of data and gave final approval of the manuscript. N.S.L., K.S.B. and C.D.C. made substantial contributions to the design of the research, analysis and interpretation of the data, manuscript drafts and final approval of the manuscript. All the authors agree to be accountable for all aspects of the work in ensuring that questions related to the accuracy or integrity of any part of the work are appropriately investigated and resolved.

Competing interests. C.D.C. is a member of the Royal Society Open Science editorial board but had no involvement in the peer review process of this submission. The authors declare no other competing interests.

Funding. The development of this manuscript and related materials by the corresponding author was in part supported by the Economic and Social Research Council (Postdoctoral Fellowship awarded to L.T.—ES/V011030/1). This research was further supported by grants held by C.D.C. from the Biotechnology and Biological Sciences Research Council (BB/K008277/1) and the European Research Council (Consolidator grant 647893 CCT).

Acknowledgements. This research project was conducted as part of the GW4 Undergraduate Psychology Consortium 2017/2018. We gratefully acknowledge Teaching Development Funding, from the faculty of Humanities and Social Sciences at the University of Bath for funding travel and room hire costs for the consortium meetings.

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
