## [Peer Review File · Royal Society Open Science]

Review History

RSOS-210666.R0 (Original submission)

Review form: Reviewer 1

Is the manuscript scientifically sound in its present form?

Yes

Are the interpretations and conclusions justified by the results?

Yes

Is the language acceptable?

Yes

Do you have any ethical concerns with this paper?

No

Have you any concerns about statistical analyses in this paper?

No

Recommendation?

Accept with minor revision (please list in comments)

Comments to the Author(s)

In this manuscript, Tzavella et al. reported a preregistered study in which they examined the effects of food-related inhibitory control training on action tendencies, liking and choice. I find the research question interesting and theoretically important, and the study design adequate for addressing the question. I also want to commend the authors for their rigor and transparency in the execution of the study, such as having a large sample size to achieve sufficient statistical power, preregistering the study, clearly distinguishing between confirmatory and exploratory analyses, sharing (well-documented) raw data and analysis code etc. The manuscript makes a meaningful contribution to the literature on ICT, and I recommend its publication with some minor revisions.

One conceptual question I have is whether the 'approach bias' as described by the BSI theory is the same as the one measured by approach-avoidance tasks. Although the BSI theory indeed talked about approach motivation toward appetitive foods, in the context of ICT, to approach specifically means to make a 'go' response (e.g., press a key), while no-go means to withhold one's response. In contrast, in approach-avoidance tasks participants always need to respond, with different responses (to pull vs. to push) designating either approach or avoidance. From this perspective, ICT may be targeting an earlier stage of action execution (whether to respond or not) than AAT (which response to execute), and as such, does not have a direct effect on the 'approach bias' as measured by AAT. I am curious to hear the authors' thoughts on this idea.

Abstract: The finding that ICT did not change approach bias towards trained foods must be taken with a grain of salt, due to the methodological issues with the AAT. As such, I believe it is important to more explicitly note that there were methodological issues with the AAT, and the null finding must be interpreted with caution in the abstract already.

Page 3, line 56: Repetition in "A total of 255 participants were recruited in total...".

Page 4: The recruited sample size (255) exceeded the sample size based on the power analysis (149), which makes me wonder what the stopping rule was when recruiting participants.

Page 6: For the AAT, participants needed to move the cursor to the top or bottom edge of the screen. I wonder over how much distance participants in general needed to move the computer mouse, and whether that required large arm flexion and extension movements or not. I'm asking this because on my own computer, I can move the mouse cursor from top to bottom with relatively small movements, which are not really comparable to when I make 'approach' and 'avoidance' movements with a joystick.

Page 8, line 26: Lee and Wagenmakers (2013) included a table in their book (page 105) on how to interpret Bayes factors. According to their table, a BF between 3 and 10 would provide moderate evidence for the alternative hypothesis. The authors however used 6 as a cut-off value for BFs. Can this be a mistake?

Page 11, Table 2: I wonder what cut-off values were used to interpret the BFs as moderate, strong, extreme etc.

Page 16, line 4: "We should also mention that a total of 50 participants were excluded due to high error rates in the AAT (> 25%)." Do the authors have explanations for why so many participants had such high error rates in the AAT?

Review form: Reviewer 2 (Eleanor Miles)

Is the manuscript scientifically sound in its present form?

Yes

Are the interpretations and conclusions justified by the results?

No

Is the language acceptable?

Yes

Do you have any ethical concerns with this paper?

No

Have you any concerns about statistical analyses in this paper?

No

Recommendation?

Accept with minor revision (please list in comments)

Comments to the Author(s)

I appreciate the transparency in this paper with regards to methods and analyses and generally felt it was a well-executed and well-written piece of research. However, I had some concerns about the validity of the ICT and AAT measures, which ultimately meant that I was unsure whether the data could be used to draw conclusions about the impact of ICT on AAT for food. If the authors could provide some more supporting evidence for the design and validity of these tasks this would allay my concerns and strengthen their conclusions.

Concerns with the ICT task: In my understanding, inhibitory control tasks should be designed to ensure that participants are actually inhibiting an already-initiated response on the no-go trials. For example, no-go trials should be less frequent than go trials so that it is beneficial for participants to initiate a response on every trial, or the no-go signal should appear after stimulus onset once a motor response has already been initiated. My understanding of the current training task is that unhealthy foods never required a response and that this was clear from the start of the trial (although, I am not 100% sure I have correctly interpreted the trial structure of the tasks, and would have found a figure useful). For these reasons, I am not convinced this task successfully trained inhibitory control. The fact that participants learned to perform this task correctly (i.e., learned not to respond to unhealthy foods) does not demonstrate that inhibitory processes were recruited, and neither do responses to the food choice task (which could be influenced by other processes).

Concerns with the AAT task: I similarly wondered whether we could be sure that the AAT was truly measuring approach and avoidance tendencies. This version of the AAT used mouse responses rather than joystick responses, which is not a trivial change to the task, given that the motor movements are quite different (I also wondered how this was implemented for online participants, e.g. how this worked for participants using trackpads). Is there any evidence for the validity of this version of the AAT? If the mouse-AAT is not a valid measure of approach and avoidance tendencies, this would explain the unexpected lack of AAT bias for unhealthy foods at pre-test. The authors do acknowledge this issue, and mention another study which failed to find pre-test differences or post-test effect using mouse responses, but this made me more concerned about the validity of the task rather than less concerned.

Given these two concerns, I was unsure how to interpret the results. It seems possible to me that the observed null effects of ICT on AAT could reflect true null effects, but that they could also reflect a failure to train ICT effectively, or a failure to measure AAT effectively. If the authors could discuss evidence to support the validity of these specific methodologies, it would strengthen their interpretation of the findings as a true null effect.

Review form: Reviewer 3

Is the manuscript scientifically sound in its present form?

No

Are the interpretations and conclusions justified by the results?

Yes

Is the language acceptable?

Yes

Do you have any ethical concerns with this paper?

No

Have you any concerns about statistical analyses in this paper?

No

Recommendation?

Accept with minor revision (please list in comments)

Comments to the Author(s)

This MS details a single study conducted as a series of parallel undergraduate research projects across three different UK universities that explored further the extent to which inhibitory control training (ICT) based on food images altered liking for, and approach towards, those images. The study complied with the highest standards of Open Science which was commendable. The outcome was somewhat underwhelming in terms of understanding how ICT might work: neither of the key hypotheses tested were supported, and that leaves the outcome inconclusive.

I have a few concerns about the way the study was conducted and interpreted, and which need noting as limitations. Firstly, although the study talks about liking for food, at no time did any participant taste or rate any food: they rated their liking for foods based on the images used in the study. Rated liking for foods based on images are effectively ratings of expected liking: we anticipate how foods will "taste" based on the images, but while liking for the appearance of food can modify actual liking for the perceived flavours, ultimately liking for flavour is the driver of intake. The way liking is presented and discussed needs to be modified to make clear that what is being rated is the expectation of flavour not actual flavour. Thus although liking for the images was unchanged, that does not preclude a possible effect of ICT on actual liking for foods when experienced.

Secondly, the study used two types of foods (savory versus sweet) and that was a key component of the study design (so some people trained to be inhibited on sweet, some on savory). The data analysis assumes that ICT will have equivalent effects on both types of food, but that is an assumption that could be tested: since savory and sweet foods play different roles in human appetite control, it would be possible to develop hypotheses why these may not be altered to the same extent by ICT.

Thirdly, the authors do not say at what time of day participants were tested. They do ask people to restrict intake before the test, but actual time of day was uncontrolled. The concern here is with the food choice task: the acceptability of some of these foods would be expected to vary considerably with time of day, and some groups who uses similar choice tasks to assess wanting versus liking responses have shown that time of day and level of hunger both modify the relative preference for savoury relative to sweet foods. Randomisation of participants to the different food-ICT conditions (ie whether sweet or savoury foods were inhibited) should have mitigated these effects to some extent but I still feel this issue needs some discussion and noted as a limitation.

Finally, the authors assume that all participants would respond similarly to ICT, but again a more nuanced account is needed since it would be predicted that those individuals who exhibit higher dietary restraint and external control would be more affected, and those who regulate their appetite based on interoceptive hunger cues less affected, by ICT. I fully recognise that including those factors in this study would make the study unwieldy, but given that lack of clear outcomes do wonder whether this was because of predictably different effects in different individuals which averaged out to a null response.

Minor comments:

Introduction line 50 "hard-wired connections" needs to be more qualified: I presume the authors mean that the neural connections between neural networks underlying Pavlovian responses and motor actions become strengthened

P5 l10-11. Most people do not know their own BMI: how was this exclusion achieved in practice (given that some data were entirely online)?

P8 l41. VAS are typically 0-100, as was used for liking. Give the rationale for 1-9 for hunger? And was this really VAS or likert?

P10 l43-47. I read this section multiple times but could not follow the explanation. This also raised some concerns about the validity of comparing these data between a multiple-user site (where there would be scope for interference between participants), individually tested in a lab and tested online.

Review form: Reviewer 4

Is the manuscript scientifically sound in its present form?

Yes

Are the interpretations and conclusions justified by the results?

Yes

Is the language acceptable?

Yes

Do you have any ethical concerns with this paper?

No

Have you any concerns about statistical analyses in this paper?

Yes

Recommendation?

Accept with minor revision (please list in comments)

Comments to the Author(s)

I find this to be a well-motivated, important and rigorous piece of research that will be of considerable value to the community. It is well-written, clearly presented and balanced in its appraisal. I have only two concerns, both of which I believe can be addressed.

My first concern is with respect to your use of default priors in your Bayes analyses. You are to be applauded for using Bayesian methods, however, the use of default priors simply isn't appropriate without a clear justification. It appears that there are ample examples of experiments using the same general methods from which effect sizes for the current study could be appropriately estimated. Those estimated effect sizes could then be used to provide a theoretically motivated model of H1 rather than simply taking an arbitrary default as you do. Taking a default under all circumstances amounts to assuming that all theories make the same predictions in all scientific contexts, which is patently false. I feel you should either explain why the specific default values used here are relevant to your specific research question, or adopt a different model of H1 based on values which can be justified.

My second concern is with the results arising from the "impulsive food choice" measure. It seems extremely likely that this task would be subject to demand characteristics i.e. it is unlikely that participants having completed the training are not aware that there is an expectation that they would be less likely to select the "no-go" associated items. The issue of demand characteristics in this work and psychological research in general is of substantial concern. The fact that the only evidence you have for an effect of training arises from the measure most obviously susceptible to this kind of influence is especially worrying. I don't believe that there is any analytical remedy and as such feel that the only thing to do is to discuss the issue of demand characteristics, and its relevance in particular to this measure, as one of the limitations in the general discussion.

Decision letter (RSOS-210666.R0)

Dear Dr Tzavella

On behalf of the Editors, we are pleased to inform you that your Manuscript RSOS-210666 "Effects of inhibitory control training on food-related action tendencies, liking and choice" has been accepted for publication in Royal Society Open Science subject to minor revision in accordance with the referees' reports. Please find the referees' comments along with any feedback from the Editors below my signature.

Please submit your revised manuscript and required files (see below) no later than 7 days from today's (ie 28-Jun-2021) date. Note: the ScholarOne system will 'lock' if submission of the revision is attempted 7 or more days after the deadline. If you do not think you will be able to meet this deadline please contact the editorial office immediately.

on behalf of Professor Zoltan Dienes (Associate Editor) and Essi Viding (Subject Editor)
openscience@royalsociety.org

Associate Editor Comments to Author (Professor Zoltan Dienes):

I invited four reviewers; and all four accepted, each of them making excellent points. All four praised the paper; indeed, I found it very well structured, with clear hypotheses and well mapped statistical tests to those hypotheses. All four make penetrating observations which I urge you to deal with fully in your response.

On the methodological side, reviewers 1 and 2 query your version of the AAT task: Given you do not use a joy stick as per the classic use of the paradigm, does the paradigm measure what it claims to? What is missing here is an outcome neutral test showing the AAT task measures what it claims. Is it worrying that there is no evidence for it indicating a preference for food overall? Maybe this relates to reviewer's 3 point about time of day and its important role in the desirability of food. Given we have no sense of what the AAT task measures, and so of what effect size it might produce if it were to detect one, I also raise Reviewer's 4 point about using default BFs. One can only obtain evidence for something not being there if there are grounds for saying what size it would be if it were to be there. This problem is not solved by using someone's default model of H1; that is because the evidence for something not being there is only as good as the grounds for claiming the effect, should it be there, is of the size modeled in the model of H1. And you have presented no grounds for saying the effect, should it be there, would be roughly around $\sqrt{2}/2$ in Cohen d/z units. (Note for example if you had used four times the number of trials in principle the expected effect size could be two times higher. But nowhere is the number of trials taken in to account in how you have modeled H1.) Now this problem might be solved if an outcome neutral test showed the AAT did measure approach vs avoidance to food when desirable, because we would have a sense of what the scale meant (for example maybe it would be relevant to use a "basic effect heuristic" as I have described in Dienes 2019 AMPPS). What should you do? The minimum I require is that, following for example reviewer 1, is that you simply qualify claims about the AAT, including in the abstract, about whether it shows what it is meant to show. Also provide some justification for why your model of H1 for BFs is justified in this scientific context for each of your DV; or else justify another model. But if you did collect data

to estimate the the effect the AAT would produce for food known to be desirable, stronger claims could be made, including a better motivated analysis (specifically the models of H1 used). Reviewer 2 also wonders if the ICT manipulation does what it is meant to. This seems an important point.

Reviewer 4 raises a plausible reinterpretation of your results: the demand characteristics for how the go/no go responses relate to food choice and liking are probably clear and probably less clear for AAT. Acknowledge this interpretation in your discussion.

Do respond to the other points I have not specifically mentioned as well.

Reviewer comments to Author:

Reviewer: 1

Comments to the Author(s)

In this manuscript, Tzavella et al. reported a preregistered study in which they examined the effects of food-related inhibitory control training on action tendencies, liking and choice. I find the research question interesting and theoretically important, and the study design adequate for addressing the question. I also want to commend the authors for their rigor and transparency in the execution of the study, such as having a large sample size to achieve sufficient statistical power, preregistering the study, clearly distinguishing between confirmatory and exploratory analyses, sharing (well-documented) raw data and analysis code etc. The manuscript makes a meaningful contribution to the literature on ICT, and I recommend its publication with some minor revisions.

One conceptual question I have is whether the 'approach bias' as described by the BSI theory is the same as the one measured by approach-avoidance tasks. Although the BSI theory indeed talked about approach motivation toward appetitive foods, in the context of ICT, to approach specifically means to make a 'go' response (e.g., press a key), while no-go means to withhold one's response. In contrast, in approach-avoidance tasks participants always need to respond, with different responses (to pull vs. to push) designating either approach or avoidance. From this perspective, ICT may be targeting an earlier stage of action execution (whether to respond or not) than AAT (which response to execute), and as such, does not have a direct effect on the 'approach bias' as measured by AAT. I am curious to hear the authors' thoughts on this idea.

Abstract: The finding that ICT did not change approach bias towards trained foods must be taken with a grain of salt, due to the methodological issues with the AAT. As such, I believe it is important to more explicitly note that there were methodological issues with the AAT, and the null finding must be interpreted with caution in the abstract already.

Page 3, line 56: Repetition in "A total of 255 participants were recruited in total...".

Page 4: The recruited sample size (255) exceeded the sample size based on the power analysis (149), which makes me wonder what the stopping rule was when recruiting participants.

Page 6: For the AAT, participants needed to move the cursor to the top or bottom edge of the screen. I wonder over how much distance participants in general needed to move the computer mouse, and whether that required large arm flexion and extension movements or not. I'm asking this because on my own computer, I can move the mouse cursor from top to bottom with relatively small movements, which are not really comparable to when I make 'approach' and 'avoidance' movements with a joystick.

Page 8, line 26: Lee and Wagenmakers (2013) included a table in their book (page 105) on how to interpret Bayes factors. According to their table, a BF between 3 and 10 would provide moderate evidence for the alternative hypothesis. The authors however used 6 as a cut-off value for BFs. Can this be a mistake?

Page 11, Table 2: I wonder what cut-off values were used to interpret the BFs as moderate, strong, extreme etc.

Page 16, line 4: "We should also mention that a total of 50 participants were excluded due to high error rates in the AAT (> 25%)." Do the authors have explanations for why so many participants had such high error rates in the AAT?

Reviewer: 2

Comments to the Author(s)

I appreciate the transparency in this paper with regards to methods and analyses and generally felt it was a well-executed and well-written piece of research. However, I had some concerns about the validity of the ICT and AAT measures, which ultimately meant that I was unsure whether the data could be used to draw conclusions about the impact of ICT on AAT for food. If the authors could provide some more supporting evidence for the design and validity of these tasks this would allay my concerns and strengthen their conclusions.

Concerns with the ICT task: In my understanding, inhibitory control tasks should be designed to ensure that participants are actually inhibiting an already-initiated response on the no-go trials. For example, no-go trials should be less frequent than go trials so that it is beneficial for participants to initiate a response on every trial, or the no-go signal should appear after stimulus onset once a motor response has already been initiated. My understanding of the current training task is that unhealthy foods never required a response and that this was clear from the start of the trial (although, I am not 100% sure I have correctly interpreted the trial structure of the tasks, and would have found a figure useful). For these reasons, I am not convinced this task successfully trained inhibitory control. The fact that participants learned to perform this task correctly (i.e., learned not to respond to unhealthy foods) does not demonstrate that inhibitory processes were recruited, and neither do responses to the food choice task (which could be influenced by other processes).

Concerns with the AAT task: I similarly wondered whether we could be sure that the AAT was truly measuring approach and avoidance tendencies. This version of the AAT used mouse responses rather than joystick responses, which is not a trivial change to the task, given that the motor movements are quite different (I also wondered how this was implemented for online participants, e.g. how this worked for participants using trackpads). Is there any evidence for the validity of this version of the AAT? If the mouse-AAT is not a valid measure of approach and avoidance tendencies, this would explain the unexpected lack of AAT bias for unhealthy foods at pre-test. The authors do acknowledge this issue, and mention another study which failed to find pre-test differences or post-test effect using mouse responses, but this made me more concerned about the validity of the task rather than less concerned.

Given these two concerns, I was unsure how to interpret the results. It seems possible to me that the observed null effects of ICT on AAT could reflect true null effects, but that they could also reflect a failure to train ICT effectively, or a failure to measure AAT effectively. If the authors could discuss evidence to support the validity of these specific methodologies, it would strengthen their interpretation of the findings as a true null effect.

Reviewer: 3

Comments to the Author(s)

This MS details a single study conducted as a series of parallel undergraduate research projects across three different UK universities that explored further the extent to which inhibitory control

training (ICT) based on food images altered liking for, and approach towards, those images. The study complied with the highest standards of Open Science which was commendable. The outcome was somewhat underwhelming in terms of understanding how ICT might work: neither of the key hypotheses tested were supported, and that leaves the outcome inconclusive.

I have a few concerns about the way the study was conducted and interpreted, and which need noting as limitations. Firstly, although the study talks about liking for food, at no time did any participant taste or rate any food: they rated their liking for foods based on the images used in the study. Rated liking for foods based on images are effectively ratings of expected liking: we anticipate how foods will “taste” based on the images, but while liking for the appearance of food can modify actual liking for the perceived flavours, ultimately liking for flavour is the driver of intake. The way liking is presented and discussed needs to be modified to make clear that what is being rated is the expectation of flavour not actual flavour. Thus although liking for the images was unchanged, that does not preclude a possible effect of ICT on actual liking for foods when experienced.

Secondly, the study used two types of foods (savory versus sweet) and that was a key component of the study design (so some people trained to be inhibited on sweet, some on savory). The data analysis assumes that ICT will have equivalent effects on both types of food, but that is an assumption that could be tested: since savory and sweet foods play different roles in human appetite control, it would be possible to develop hypotheses why these may not be altered to the same extent by ICT.

Thirdly, the authors do not say at what time of day participants were tested. They do ask people to restrict intake before the test, but actual time of day was uncontrolled. The concern here is with the food choice task: the acceptability of some of these foods would be expected to vary considerably with time of day, and some groups who uses similar choice tasks to assess wanting versus liking responses have shown that time of day and level of hunger both modify the relative preference for savory relative to sweet foods. Randomisation of participants to the different food-ICT conditions (ie whether sweet or savory foods were inhibited) should have mitigated these effects to some extent but I still feel this issue needs some discussion and noted as a limitation.

Finally, the authors assume that all participants would respond similarly to ICT, but again a more nuanced account is needed since it would be predicted that those individuals who exhibit higher dietary restraint and external control would be more affected, and those who regulate their appetite based on interoceptive hunger cues less affected, by ICT. I fully recognise that including those factors in this study would make the study unwieldy, but given that lack of clear outcomes do wonder whether this was because of predictably different effects in different individuals which averaged out to a null response.

Minor comments:

Introduction line 50 “hard-wired connections” needs to be more qualified: I presume the authors mean that the neural connections between neural networks underlying Pavlovian responses and motor actions become strengthened

P5 l10-11. Most people do not know their own BMI: how was this exclusion achieved in practice (given that some data were entirely online)?

P8 l41. VAS are typically 0-100, as was used for liking. Give the rationale for 1-9 for hunger? And was this really VAS or likert?

P10 143-47. I read this section multiple times but could not follow the explanation. This also raised some concerns about the validity of comparing these data between a multiple-user site (where there would be scope for interference between participants), individually tested in a lab and tested online.

Reviewer: 4

Comments to the Author(s)

I find this to be a well-motivated, important and rigorous piece of research that will be of considerable value to the community. It is well-written, clearly presented and balanced in its appraisal. I have only two concerns, both of which I believe can be addressed.

My first concern is with respect to your use of default priors in your Bayes analyses. You are to be applauded for using Bayesian methods, however, the use of default priors simply isn't appropriate without a clear justification. It appears that there are ample examples of experiments using the same general methods from which effect sizes for the current study could be appropriately estimated. Those estimated effect sizes could then be used to provide a theoretically motivated model of H1 rather than simply taking an arbitrary default as you do.

Taking a default under all circumstances amounts to assuming that all theories make the same predictions in all scientific contexts, which is patently false. I feel you should either explain why the specific default values used here are relevant to your specific research question, or adopt a different model of H1 based on values which can be justified.

My second concern is with the results arising from the "impulsive food choice" measure. It seems extremely likely that this task would be subject to demand characteristics i.e. it is unlikely that participants having completed the training are not aware that there is an expectation that they would be less likely to select the "no-go" associated items. The issue of demand characteristics in this work and psychological research in general is of substantial concern. The fact that the only evidence you have for an effect of training arises from the measure most obviously susceptible to this kind of influence is especially worrying. I don't believe that there is any analytical remedy and as such feel that the only thing to do is to discuss the issue of demand characteristics, and its relevance in particular to this measure, as one of the limitations in the general discussion.

===PREPARING YOUR MANUSCRIPT===

While not essential, it will speed up the preparation of your manuscript proof if you format your references/bibliography in Vancouver style (please see

<https://royalsociety.org/journals/authors/author-guidelines/#formatting>). You should include DOIs for as many of the references as possible.

===PREPARING YOUR REVISION IN SCHOLARONE===

<https://royalsociety.org/journals/authors/author-guidelines/#data>. You should ensure that you cite the dataset in your reference list. If you have deposited data etc in the Dryad repository,

please only include the 'For publication' link at this stage. You should remove the 'For review' link.

Author's Response to Decision Letter for (RSOS-210666.R0)

See Appendix A.

Decision letter (RSOS-210666.R1)

Dear Dr Tzavella,

It is a pleasure to accept your manuscript entitled "Effects of go/no-go training on food-related action tendencies, liking and choice" in its current form for publication in Royal Society Open Science. The comments of the Editors are included at the foot of this letter.

on behalf of Professor Zoltan Dienes (Associate Editor) and Essi Viding (Subject Editor)
openscience@royalsociety.org

Associate Editor Comments to Author (Professor Zoltan Dienes):

I think you have responded well to all points.

You still use something like a default for your additional BF analyses, whereas for the BFs to strictly apply to the theory tested one has to show the prior represents the predictions of that very theory. However, I think your argument that with high N it won't matter very much in this case is acceptable. (For next time, showing the *range* of scale factors for which a conclusion holds can help readers who have their arguments for a particular scale factor.)

Your paper brings together several aspects of good practice not routinely followed in the literature, and I have to say illustrates an excellent training of undergraduates.

Appendix A

Response to Associate Editor's and Reviewers' comments

Associate Editor Comments to Author (Professor Zoltan Dienes):

I invited four reviewers; and all four accepted, each of them making excellent points. All four praised the paper; indeed, I found it very well structured, with clear hypotheses and well mapped statistical tests to those hypotheses. All four make penetrating observations which I urge you to deal with fully in your response.

On the methodological side, reviewers 1 and 2 query your version of the AAT task: Given you do not use a joy stick as per the classic use of the paradigm, does the paradigm measure what it claims to? What is missing here is an outcome neutral test showing the AAT task measures what it claims. Is it worrying that there is no evidence for it indicating a preference for food overall? Maybe this relates to reviewer's 3 point about time of day and its important role in the desirability of food. Given we have no sense of what the AAT task measures, and so of what effect size it might produce it if were to detect one, I also raise Reviewer's 4 point about using default BFs. One can only obtain evidence for something not being there if there are grounds for saying what size it would be if it were to be there. This problem is not solved by using someone's default model of H1; that is because the evidence for something not being there is only as good as the grounds for claiming the effect, should it be there, is of the size modeled in the model of H1. And you have presented no grounds for saying the effect, should it be there, would be roughly around $\sqrt{2}/2$ in Cohen d/z units. (Note for example if you had used four times the number of trials in principle the expected effect size could be two times higher. But nowhere is the number of trials taken in to account in how you have modeled H1.) Now this problem might be solved if an outcome neutral test showed the AAT did measure approach vs avoidance to food when desirable, because we would have a sense of what the scale meant (for example maybe it would be relevant to use a "basic effect heuristic" as I have described in Dienes 2019 AMPPS). What should you do? **The minimum I require is that, following for example reviewer 1, is that you simply qualify claims about the AAT, including in the abstract, about whether it shows what it is meant to show. Also provide some justification for why your model of H1 for BFs is justified in this scientific context for each of your DV; or else justify another model.** But if you did collect data to estimate the effect the AAT would produce for food known to be desirable, stronger claims could be made, including a better motivated analysis (specifically the models of H1 used).

Reviewer 2 also wonders if the ICT manipulation does what it is meant to. This seems an important point.

Reviewer 4 raises a plausible reinterpretation of your results: the demand characteristics for how the go/no go responses relate to food choice and liking are probably clear and probably less clear for AAT. Acknowledge this interpretation in your discussion.

Do respond to the other points I have not specifically mentioned as well.

Reply: Many thanks for the careful editorial guidance. We have updated our abstract to highlight the potential limitations of the AAT as Reviewer 1 suggested. Please note the first two sentences have been altered as well to meet the 200-word limit. The conceptual issue raised by Reviewer 2 was addressed by adding a new paragraph to our introduction and updating our main text where appropriate to specifically mention 'go/no-go training' as opposed to the broader term of 'inhibitory control training'. We have also added supplementary analyses with an adjusted prior in the Supplementary Info to highlight the issue of prior specification and these results are presented as a robustness check as the interpretation of outcomes did not change. However, we now present a table that shows how the strength of evidence for H1/H0 changes with the choice of a prior (p3). For these supplementary analyses we also provide the corresponding JASP plots so that readers can examine the posterior distributions as well (p4).

Reviewer comments to Author:

Reviewer: 1

Comments to the Author(s)

In this manuscript, Tzavella et al. reported a preregistered study in which they examined the effects of food-related inhibitory control training on action tendencies, liking and choice. I find the research question interesting and theoretically important, and the study design adequate for addressing the question. I also want to commend the authors for their rigor and transparency in the execution of the study, such as having a large sample size to achieve sufficient statistical power, preregistering the study, clearly distinguishing between confirmatory and exploratory analyses, sharing (well-documented) raw data and analysis code etc. The manuscript makes a meaningful contribution to the literature on ICT, and I recommend its publication with some minor revisions.

One conceptual question I have is whether the 'approach bias' as described by the BSI theory is the same as the one measured by approach-avoidance tasks. Although the BSI theory indeed talked about approach motivation toward appetitive foods, in the context of ICT, to approach specifically means to make a 'go' response (e.g., press a key), while no-go means to withhold one's response. In contrast, in approach-avoidance tasks participants always need to respond, with different responses (to pull vs. to push) designating

either approach or avoidance. From this perspective, ICT may be targeting an earlier stage of action execution (whether to respond or not) than AAT (which response to execute), and as such, does not have a direct effect on the 'approach bias' as measured by AAT. I am curious to hear the authors' thoughts on this idea.

Reply: We are very thankful for these comments and agree with Reviewer 1 there may indeed be conceptual issues that require further formal investigation in future studies. We had considered that difference as well and for this reason we also looked into action initiation times in the AAT, rather than completion times which include response execution time. However, the results were still the same and there was no evidence for approach bias as measured by that version of the AAT. Of course, there is not enough evidence to suggest that action initiation in a task such as the AAT is consistent with a go response in GNG and that this measure is sensitive enough to capture earlier stages of action. This is one of the reasons we believe that more research is required to provide evidence for such links that are often mentioned in prominent theoretical frameworks for ICT effects.

We have added the following sentence to the Conclusion to further highlight this point for future research:

"The issue of operationalisation in this research area may require more empirical attention as it would be of theoretical and applied interest to know whether go responses and approach motivation towards foods in the GNG [e.g., see BSI theory; 33] can be directly mapped onto automatic action tendencies, as measured by other experimental tasks in the literature."

Abstract: The finding that ICT did not change approach bias towards trained foods must be taken with a grain of salt, due to the methodological issues with the AAT. As such, I believe it is important to more explicitly note that there were methodological issues with the AAT, and the null finding must be interpreted with caution in the abstract already.

Reply: We have updated our abstract to explicitly note that as you suggested: "...we warrant caution in interpreting this finding as there are important limitations to consider for the employed approach-avoidance task."

Page 3, line 56: Repetition in "A total of 255 participants were recruited in total..."

Reply: The repetition has been corrected in the main text.

Page 4: The recruited sample size (255) exceeded the sample size based on the power analysis (149), which makes me wonder what the stopping rule was when recruiting participants.

Reply: We thank the reviewer for identifying this issue with the manuscript, as we hoped to include as much information as possible regarding our data collection procedure to maximize transparency. We have now provided an explanation of our stopping rule in the main text, including a note on how and why we ended up having more participants in our sample (see section 5.1. Sample Characteristics). Please note that the *recruited* sample size is not the same as the *target* sample size as it does not account for data exclusions. Our target sample size was 149 and we exceeded that by only 14 participants.

Page 6: For the AAT, participants needed to move the cursor to the top or bottom edge of the screen. I wonder over how much distance participants in general needed to move the computer mouse, and whether that required large arm flexion and extension movements or not. I'm asking this because on my own computer, I can move the mouse cursor from top to bottom with relatively small movements, which are not really comparable to when I make 'approach' and 'avoidance' movements with a joystick.

Reply: We agree with this observation and did consider this issue when programming the task and collecting the data. The distance that participants had to travel with the mouse was greater due to the selected settings of the mouse speed. We added a clarification for this in our Supplementary Information (S3. AAT Responses), and the text is also copied below:

"The laboratory testing guidelines for data collection across sites included an important step for ensuring that the AAT can be performed appropriately. Researchers were asked to set the exact speed of the mouse before starting the study and also disable the default acceleration of the cursor during movement initiation. Reducing the speed of the mouse (4th notch from the left in Windows-based settings) meant that participants did not complete the whole-arm movements too fast for 'pull' and 'push' actions."

Page 8, line 26: Lee and Wagenmakers (2013) included a table in their book (page 105) on how to interpret Bayes factors. According to their table, a BF between 3 and 10 would provide moderate evidence for the alternative hypothesis. The authors however used 6 as a cut-off value for BFs. Can this be a mistake?

Reply: Thank you for spotting this. We follow the guidelines by Lee & Wagenmakers and this cut-off was adapted from sampling plan stopping rules where instead of a BF of 10 you choose a less conservative BF of 6 - yet not 3 which is closer to anecdotal evidence. Indeed this is an error with regards to how evidence strength is interpreted in the main paper and it has been corrected.

Page 11, Table 2: I wonder what cut-off values were used to interpret the BFs as moderate, strong, extreme etc.

Reply: We used the cut-off values from Lee & Wagenmakers (2013) and we added a note on the table caption to indicate where this information can be found (section 4.2).

Page 16, line 4: "We should also mention that a total of 50 participants were excluded due to high error rates in the AAT (> 25%)." Do the authors have explanations for why so many participants had such high error rates in the AAT?

Reply: Unfortunately, we are not sure whether there were additional motor execution demands for some participants that led to increased task difficulty or whether there were instruction comprehension issues, or attentional issues. The researchers who collected the data in the laboratory did report that several participants took too long to start a response. In the discussion we touched upon this issue by highlighting previous research that reported similar issues (only one found). However, to address this comment we have now added another sentence in that paragraph (see underlined text below).

"Consistent with our observations about AAT performance, the authors also reported relatively high error rates compared to other studies (e.g., 11% at baseline) and indicated that the use of the computer mouse could have increased measurement error [78]. More research is required to examine the conditions under which performance in this AAT variant can be improved in terms of accuracy and arm movements."

Reviewer: 2

Comments to the Author(s)

I appreciate the transparency in this paper with regards to methods and analyses and generally felt it was a well-executed and well-written piece of research. However, I had some concerns about the validity of the ICT and AAT measures, which ultimately meant that I was unsure whether the data could be used to draw conclusions about the impact of ICT on AAT for food. If the authors could provide some more supporting evidence for the design and validity of these tasks this would allay my concerns and strengthen their conclusions.

Concerns with the ICT task: In my understanding, inhibitory control tasks should be designed to ensure that participants are actually inhibiting an already-initiated response on the no-go trials. For example, no-go trials should be less frequent than go trials so that it is beneficial for participants to initiate a response on every trial, or the no-go signal should appear after stimulus onset once a motor response has already been initiated. My understanding of the current training task is that unhealthy foods never required a response and that this was clear from the start of the trial (although, I am not 100% sure I have correctly interpreted the trial structure of the tasks, and would have found a figure useful). For these reasons, I am not convinced this task successfully trained inhibitory control.

Reply: Thank you for raising this interesting point. The question of whether go/no-go training (GNG) taps into 'inhibitory control' is important as the two paradigms used in the literature vary in the type of inhibition that is required by participants - e.g., reactive vs proactive inhibition, or action cancellation vs action restraint. In Figure 1, panel B we show both the trial structure in the GNG and the assignment of foods to different trials. All the foods included in this paradigm were energy-dense ('unhealthy') but specific foods were randomly matched with different types of trials. **Go** foods were always presented on go trials, while **no-go** foods were always paired with a signal. The **control** or filler foods were not consistently associated with go/no-go responses as they were presented with and without a signal (50%, 50%). Please note that this is also included in the abstract.

We agree that this is an important conceptual issue that needs to be highlighted more in the manuscript and for this reason we have updated the manuscript to avoid the term 'inhibitory control training' when discussing the specific paradigm we used- although GNG training is included in the ICT literature - we consider this point critical enough to adapt our main text accordingly. Specifically, we have updated our title and added a paragraph in the introduction to explain these issues, including a footnote that discloses why we use the term and how it can be interpreted. For convenience I have copied both below.

"Although in both ICT paradigms participants are required to inhibit responses towards target stimuli, the type of response inhibition that may be required is not necessarily the same. In go/no-go training participants are presented with a cue/signal on 50% of the trials with zero-to-little delay, while in stop-signal training tasks the signal is only shown on a minority of trials and its

onset is delayed to maintain task difficulty [see 29 for comparison]. Although go/no-go training may not necessarily tap into ‘inhibitory control’* as a top-down process, meta-analyses have shown that compared to stop-signal training it is more effective in changing eating-related behaviour [9, 10].”

*[Footnote] ”We use the term ‘inhibitory control training’ throughout this manuscript to refer to both paradigms that are commonly reported as such in the literature, but we acknowledge that there is not enough evidence to suggest that go/no-go training taps into top-down inhibitory control mechanisms, as opposed to automatic inhibition [e.g. automatic or controlled inhibition; see 39].”.

The fact that participants learned to perform this task correctly (i.e., learned not to respond to unhealthy foods) does not demonstrate that inhibitory processes were recruited, and neither do responses to the food choice task (which could be influenced by other processes).

Reply: We agree with Reviewer 2 that contingency learning does not necessarily suggest that inhibitory control processes were recruited but rather that stimulus-response (S-R) associations may have formed -this can be explained by an automatic inhibition account (i.e., this type of learning does not rely on top-down inhibitory control). This clarification has been added as mentioned in the previous reply, but for contingency learning specifically we did not make any assumptions regarding inhibitory control; e.g., see this sentence from the discussion where we present these results:

“First, the manipulation check for contingency learning during training was successful which indicates that stimulus-response associations were formed during the GNG paradigm consistent with previous literature [see 27,63].”

Concerns with the AAT task: I similarly wondered whether we could be sure that the AAT was truly measuring approach and avoidance tendencies. This version of the AAT used mouse responses rather than joystick responses, which is not a trivial change to the task, given that the motor movements are quite different (I also wondered how this was implemented for online participants, e.g. how this worked for participants using trackpads). Is there any evidence for the validity of this version of the AAT? If the mouse-AAT is not a valid measure of approach and avoidance tendencies, this would explain the unexpected lack of AAT bias for unhealthy foods at pre-test. The authors do acknowledge this issue, and mention another study which failed to find pre-test differences or post-test effect using mouse responses, but this made me more concerned about the validity of the task rather than less concerned.

Reply: We agree with these concerns about the AAT and we would like to note that the issue of trackpads and motor movements was taken into account when planning the study - the participants who did the study online had a computer mouse and they were required to change the mouse speed in line with the laboratory guidelines for testing (we added the guidelines in the Supplementary Information). With regards to the use of a computer as opposed to a joystick, there are unfortunately not enough studies that report issues of reliability, validity and measurement error as they use this variant as a training task and not an outcome task. In the discussion we have included a relevant study which reported similar issues with AAT performance and thus we agree that this variant of the AAT requires further investigation. We have also highlighted this in the discussion as part of addressing a comment by Reviewer 1 (see below; underlined text has been added for this revision).

“Consistent with our observations about AAT performance, the authors also reported relatively high error rates compared to other studies (e.g., 11% at baseline) and indicated that the use of the computer mouse could have increased measurement error [78]. More research is required to examine the conditions under which performance in this AAT variant can be improved in terms of accuracy and arm movements.”

Given these two concerns, I was unsure how to interpret the results. It seems possible to me that the observed null effects of ICT on AAT could reflect true null effects, but that they could also reflect a failure to train ICT effectively, or a failure to measure AAT effectively. If the authors could discuss evidence to support the validity of these specific methodologies, it would strengthen their interpretation of the findings as a true null effect.

Reply: As noted above, GNG tasks that train S-R associations rather than top-down ICT have been shown to be more effective in changing behaviour so we do not believe a failure to train "ICT" effectively is a likely explanation for our results. Similarly, the specific paradigm has been adapted from previous literature that has reported effects on behaviour and our results on the food choice and liking measures suggest otherwise (i.e., training worked as expected; devaluation effect is smaller however, at $d = -0.19$). Please note that we have added a supplementary analysis for these tests with tailored priors, as part of this revision, that do provide moderate, instead of anecdotal, evidence for a devaluation effect. However, we agree that there are reasons to be concerned with the validity of the approach bias measure, which has been noted in the manuscript. Considering that this is a ‘novel’ study design that uses the AAT as an outcome measure - and there are potential limitations outlined in the discussion for this - it is not possible to infer whether this a ‘true’ null effect as operationalisation of action tendencies may be problematic in this context. Even with

what we consider well-established measures in psychology there are known problems with reliability and validity - and this is why we believe the exploratory analyses (reliability) and discussed limitations are important for future work. The abstract has been revised to account for that uncertainty/attention on the interpretation of findings.

Please note that we also highlight this in our Conclusion: “Nevertheless, there are potential methodological limitations regarding the design of the AAT as an indirect measure of motivational bias in this context that need to be addressed before drawing any conclusions.”

Reviewer: 3

Comments to the Author(s)

This MS details a single study conducted as a series of parallel undergraduate research projects across three different UK universities that explored further the extent to which inhibitory control training (ICT) based on food images altered liking for, and approach towards, those images. The study complied with the highest standards of Open Science which was commendable. The outcome was somewhat underwhelming in terms of understanding how ICT might work: neither of the key hypotheses tested were supported, and that leaves the outcome inconclusive.

I have a few concerns about the way the study was conducted and interpreted, and which need noting as limitations. Firstly, although the study talks about liking for food, at no time did any participant taste or rate any food: they rated their liking for foods based on the images used in the study. Rated liking for foods based on images are effectively ratings of expected liking: we anticipate how foods will “taste” based on the images, but while liking for the appearance of food can modify actual liking for the perceived flavours, ultimately liking for flavour is the driver of intake. The way liking is presented and discussed needs to be modified to make clear that what is being rated is the expectation of flavour not actual flavour. Thus although liking for the images was unchanged, that does not preclude a possible effect of ICT on actual liking for foods when experienced.

Reply: We agree with this definition of liking from that perspective but in this study liking reflects a combination of ‘liking’ and ‘wanting’ (these ratings were very highly correlated in some of our previous research, e.g. Lawrence et al., 2012) and our aim here was not to clearly delineate these different processes but to effectively capture the subjective *value* of the food in an intuitive way to assess the devaluation effect of training. In response to this comment, we have added a sentence in the Discussion, as shown below.

“It may also be worth investigating whether actual food liking (rating the flavour of foods in the laboratory) would be a more critical determinant of training effects as opposed to ratings of *expected* liking/taste.”

Secondly, the study used two types of foods (savoury versus sweet) and that was a key component of the study design (so some people trained to be inhibited on sweet, some on savoury). The data analysis assumes that ICT will have equivalent effects on both types of food, but that is an assumption that could be tested: since savoury and sweet foods play different roles in human appetite control, it would be possible to develop hypotheses why these may not be altered to the same extent by ICT.

Reply: We agree with this design consideration and took this into account in our training paradigm; i.e., people trained to inhibit their responses towards sweet *and* savoury foods. This is explained in section 3.4 and the relevant text is copied below:

“Two food categories were randomly assigned to each training condition (go, no-go, control foods) in the beginning of the experiment and food taste was counterbalanced so that each condition had one sweet and one savoury food.”

Thirdly, the authors do not say at what time of day participants were tested. They do ask people to restrict intake before the test, but actual time of day was uncontrolled. The concern here is with the food choice task: the acceptability of some of these foods would be expected to vary considerably with time of day, and some groups who uses similar choice tasks to assess wanting versus liking responses have shown that time of day and level of hunger both modify the relative preference for savoury relative to sweet foods. Randomisation of participants to the different food-ICT conditions (ie whether sweet or savoury foods were inhibited) should have mitigated these effects to some extent but I still feel this issue needs some discussion and noted as a limitation.

Reply: As noted in the previous reply, the relative preference for savoury relative to sweet foods would not be a methodological issue to consider in our case because we counterbalanced taste (sweet and savoury) for each training condition. We aimed for participants not to be fully satiated but also not very hungry as responses in the outcome tasks could be affected by this (e.g., choose the food that would be most filling at that moment). Our data indicated that the majority of participants had a meal 3-5 hours before the study, thus adhering to the instruction not to eat 3 hours before, but those who did not follow this instruction were not excluded.

Overall, participants' self-report hunger levels were neither too high nor too low ($M = 5.70$ on a scale from 1 to 9). Exploratory analyses that were conducted as part of the student projects for this consortium had shown that hunger, hours since last meal and related factors did not correlate significantly with training effects on approach bias or liking.

Finally, the authors assume that all participants would respond similarly to ICT, but again a more nuanced account is needed since it would be predicted that those individuals who exhibit higher dietary restraint and external control would be more affected, and those who regulate their appetite based on interoceptive hunger cues less affected, by ICT. I fully recognise that including those factors in this study would make the study unwieldy, but given that lack of clear outcomes do wonder whether this was because of predictably different effects in different individuals which averaged out to a null response.

Reply: We agree with the reviewer that these factors could have been measured at the end of the study to allow for a more in-depth exploration of training outcomes and the extent to which approach bias for foods at baseline would vary under different conditions. We have now added this point in our Conclusion, as shown below.

“Similarly, future research could further explore the role of individual differences in training outcomes and complementary or underlying mechanisms (approach bias, stimulus devaluation) by measuring factors such as restrained and external eating [e.g., 21,23,46,86].“

Minor comments:

Introduction line 50 “hard-wired connections” needs to be more qualified: I presume the authors mean that the neural connections between neural networks underlying Pavlovian responses and motor actions become strengthened

Reply: We have added the term ‘neural’ in this sentence. This idea is further explained in the rest of the paragraph. We also included the phrase ‘increased avoidance’ to explain how no-go stimuli would be devalued when there is a mutually facilitatory connection between the no-go system and the aversive centre.

P5 l10-11. Most people do not know their own BMI: how was this exclusion achieved in practice (given that some data were entirely online)?

Reply: BMI was calculated using the self-reported height and weight of the participants; i.e. we didn't ask participants to type in their own BMI (see section 3.8).

P8 l41. VAS are typically 0-100, as was used for liking. Give the rationale for 1-9 for hunger? And was this really VAS or likert?

Reply: We agree with this observation and have therefore removed the term VAS from the text and used ‘Likert’ instead. We have consistently used this scale for other training studies as we do not believe there is a need for a more granular assessment of hunger (unless used in covariate analyses).

P10 l43-47. I read this section multiple times but could not follow the explanation. This also raised some concerns about the validity of comparing these data between a multiple-user site (where there would be scope for interference between participants), individually tested in a lab and tested online.

Reply: First, we would like to thank the reviewer for pointing out the lack of clarity in our statement regarding the data collection procedure. We have now removed this sentence and replaced it with a footnote that explains exactly why and how the final sample size included 14 more participants. Second, we do not have any concerns about the validity of the data with regards to the multi-site data collection procedure as our setup was standardised across sites in advance (computer settings, requirements, mouse speed etc.).

Reviewer: 4

Comments to the Author(s)

I find this to be a well-motivated, important and rigorous piece of research that will be of considerable value to the community. It is well-written, clearly presented and balanced in its appraisal. I have only two concerns, both of which I believe can be addressed.

My first concern is with respect to your use of default priors in your Bayes analyses. You are to be applauded for using Bayesian methods, however, the use of default priors simply isn't appropriate without a clear justification. It appears that there are ample

examples of experiments using the same general methods from which effect sizes for the current study could be appropriately estimated. Those estimated effect sizes could then be used to provide a theoretically motivated model of H1 rather than simply taking an arbitrary default as you do. Taking a default under all circumstances amounts to assuming that all theories make the same predictions in all scientific contexts, which is patently false. I feel you should either explain why the specific default values used here are relevant to your specific research question, or adopt a different model of H1 based on values which can be justified.

Reply: Thank you for the positive feedback on our manuscript and the points raised. We have addressed both issues as suggested in our revision.

First, we agree that the use of default priors is not ideal and in retrospect we could have tailored our analysis better to the expected effects of interest. Unfortunately, at the time of preregistration we did not have enough information and expertise to properly justify the choice of informative prior parameters and considered that default priors would not be a problem given our target sample size (e.g., conclusion will still be the same but strength of evidence may differ).

Our position since that time has changed and we therefore agree with this comment. As we had previously noted in our Supplementary Information (“For all Bayesian tests we used the default priors in JASP, but for potential replication and/or extensions priors tailored to small-to-medium effects are preferred [e.g., see discussion in 11]”), we do believe the prior parameters could be tailored to fit the expected effect size of interest included in the frequentist power analysis. For this reason, we have included analyses with adjusted priors (not based on the posterior distributions, but what we would have employed if we were formulating an analysis plan again with that effect size in mind).

The results are presented in the Supplementary Information (new section, S6) and we have included a table which compares the grades/strength of evidence for all hypotheses that concern training effects (H1, H2, H3). This could be deemed a robustness check for the preregistered analyses as the results were overall consistent irrespective of the prior distribution - especially with regards to the AAT effects (still evidence favours H0 over H1). As we do not wish to deviate from the preregistered protocol, we believe this is an adequate addition to the manuscript and we are very thankful for this comment as this can potentially help researchers who do similar or follow-up work to carefully consider the impact of prior parameters on their Bayesian analyses.

My second concern is with the results arising from the “impulsive food choice” measure. It seems extremely likely that this task would be subject to demand characteristics i.e. it is unlikely that participants having completed the training are not aware that there is an expectation that they would be less likely to select the “no-go” associated items. The issue of demand characteristics in this work and psychological research in general is of substantial concern. The fact that the only evidence you have for an effect of training arises from the measure most obviously susceptible to this kind of influence is especially worrying. I don’t believe that there is any analytical remedy and as such feel that the only thing to do is to discuss the issue of demand characteristics, and its relevance in particular to this measure, as one of the limitations in the general discussion.

Reply: We appreciate the feedback on our food choice task, and we agree that many measures included in this area of research are prone to demand characteristics, but we should note that previous work that included a funnelled debrief at the end of the study argues against the influence of demand characteristics (Lawrence et al. 2015; study 1 and study 2). This version of the impulsive choice task requires participants to choose foods within a strict time limit and the study only presents energy-dense foods, which may mean that participants would need to be aware of the specific stimulus-response contingencies (e.g., cake is a no-go food while crisps is a go food, instead of ‘unhealthy’ is no-go and ‘healthy’ is go). We did not have a measure of awareness for stimulus-response contingencies, but we felt that it is important to address this comment in the discussion and therefore added the paragraph below.

Recent evidence suggests that training effects may only be reliable for speeded, and not for deliberate, food choice [26,72], which indicates that demand characteristics would not affect the results in this study even if a proportion of participants was aware of stimulus-response contingencies after training (i.e., cake was a no-go food so I will not choose it). Previous research has further shown that memory of stimulus-response contingencies did not affect food choice outcomes [26]. Although the food choice task in this study required participants to respond within a time limit, future replications and/or extensions of these findings could still employ other impulsive choice measures that are less prone to strategic responding, as for example the speeded binary food choice task which involves multiple choice combinations and stricter time windows [see 85].